# MorphGen: Controllable and Morphologically Plausible Generative Cell-Imaging

## Abstract

Simulating in silico cellular responses to interventions is a promising direction to accelerate high-content image-based assays, critical for advancing drug discovery and gene editing. To support this, we introduce MorphGen, a state-of-the-art diffusion-based generative model for fluorescent microscopy that enables controllable generation across multiple cell types and perturbations. To capture biologically meaningful patterns consistent with known cellular morphologies, MorphGen is trained with an alignment loss to match its representations to the phenotypic embeddings of OpenPhenom, a state-of-the-art biological foundation model. Unlike prior approaches that compress multichannel stains into RGB images – sacrificing organelle-specific detail – MorphGen generates the complete set of fluorescent channels jointly, preserving per-organelle structures and enabling a fine-grained morphological analysis that is essential for post-generation biological interpretation. We demonstrate biological consistency with real images via CellProfiler features, and MorphGen attains an FID score over $35\%$ lower than the prior state-of-the-art MorphoDiff, which only generates RGB images for a single cell type.

## 1 Introduction

Deep generative models are emerging tools for simulating cellular behavior in computational biology, with early works in modeling gene expression profiles (Cui et al., 2024; Bereket & Karaletsos, 2024) and more recently synthesizing microscopy images (Navidi et al., 2025; Palma et al., 2025), which can be easily collected at scale. These models offer the potential to create *in silico* surrogates of biological experiments – virtual systems that capture cellular morphology and its response to perturbations. This is a critical step in the vision of *Virtual Cells* (Bunne et al., 2024): a generative instrument capable of populating diverse cellular contexts and emulating the effects of genetic or chemical interventions. Realizing such a system could accelerate biological discovery by producing high-quality hypotheses without the time and cost constraints of exhaustive wet-lab experiments. As a practical step toward this vision, we focus on phenotypic image generation under experimentally defined perturbations.

In fact, among the current experimental platforms, microscopy-based high-content screening (HCS) provides a particularly fitting setting for generative models of cellular phenotypes under different perturbations. Automated microscopes now capture hundreds of thousands of single-cell images per experiment, and image-analysis pipelines distill them into high-dimensional phenotypes that reveal subtle biological variations invisible to bulk assays (Boutros et al., 2015; Caicedo et al., 2017). These rich readouts make HCS an attractive target for *in silico* modeling, where generative models could simulate phenotypic outcomes across perturbations that would be costly or impractical to assay exhaustively.

A standout HCS protocol is Cell Painting, which stains eight cellular compartments across six fluorescence channels. The resulting morphological fingerprints have proved versatile: they cluster small molecules by mechanism of action (Wawer et al., 2014), map gene function (Rohban et al., 2017), and capture disease-specific signatures for phenotypic drug discovery (Vincent et al., 2022). With its scale and diversity, Cell Painting enable generative models to learn and generate realistic and biologically faithful images.

However, current image generators fall short of these goals: (i) they operate at low resolution and rely on outdated architectures (Palma et al., 2025); (ii) they collapse six-channel fluorescence stacks into lossy RGB, discarding biological detail (Phillips, 2025); and (iii) they are restricted to a single cell type and modest-sized datasets (Navidi et al., 2025). As a result, they miss out on both fine-grained

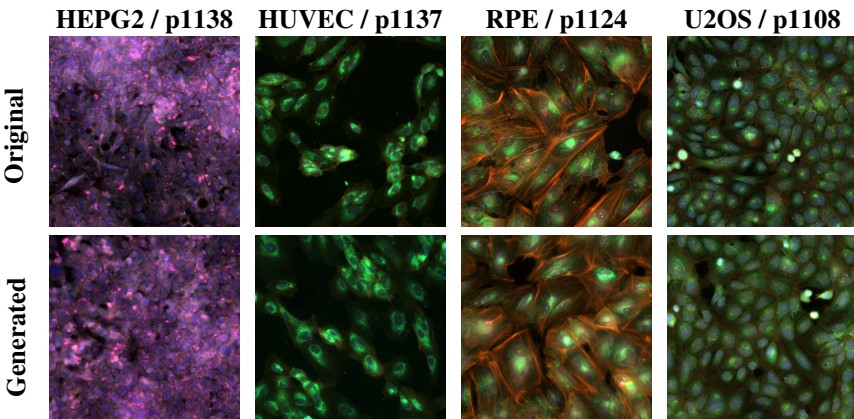

Figure 1: Original (top row) and generated (bottom row) images for various cell type / perturbation ID pairs from the RxRx1 dataset (Sypetkowski et al., 2023). Unlike existing models, our MorphGen is capable of generating crisp, high-dimensional images across different cell-types and perturbations. Generated images are not cherry-picked, and we selected original images that are neighbors of the generated ones for visualization. See Appendix Q for additional examples.

morphological analysis and realism. Instead, we posit that a generative model should maintain local biological information, even at the individual fluorescence level. Further, restricting generation to a single cell type limits the model's generality, posing a key obstacle toward applications.

We present **MorphGen**, a generative model that addresses these gaps and supports virtual phenotyping across many perturbations and four cell lines. Its design and empirical contributions are:

- **Organelle-level generation.** MorphGen models native fluorescence channels directly, avoiding RGB conversion and preserving sub-cellular detail.

- **Domain-aligned regularization.** During training, an alignment loss guided by embeddings from the biological foundation model *OpenPhenom* (Kraus et al., 2024) encourages the diffusion model to capture biologically meaningful features, leading to higher-fidelity images.

- **Flexible conditioning.** The latent space separates perturbation and cell-type factors, allowing controllable synthesis, for example, swapping cell-type while keeping the perturbation fixed.

- **Scalable training.** We train MorphGen at full resolution on the entire RxRx1 dataset (Sypetkowski et al., 2023), spanning four cell types and over 125K images of resolution $512 \times 512$.

- **Downstream validation.** Synthetic images reproduce realistic population-level statistics, which we measure as conditional average treatment effects (CATEs) of morphological features across perturbations and controls, confirming suitability for in-silico screening.

Figure 6 illustrates the visual fidelity of our model across four representative cell-type/perturbation pairs. To the best of our knowledge, MorphGen is the first generator that delivers high-resolution, organelle-aware, and biologically faithful Cell Painting images at scale for practical *in silico* screening.

## 2    RELATED WORK ON GENERATIVE MODELS IN MICROSCOPY-BASED HCS

In this section we discuss the recent work that attempts controllable generation of Cell-Painting images to illustrate the morphological response of a given perturbation. Here, we outline the essentials and provide a comparison with key design choices in MorphGen.

**Morphodiff.** MorphoDiff (Navidi et al., 2025) adapts a Stable-Diffusion (Rombach et al., 2022) latent DDPM (Ho et al., 2020) to Cell Painting. Perturbations are encoded with scGPT embeddings (Cui et al., 2024). Six fluorescence channels are projected into RGB through an irreversible compression that merges organelle-specific cues (Phillips, 2025) to remain compatible with the pretrained Stable Diffusion VAE (Kingma & Welling, 2013). The model trains on full resolution images from a single cell type in RxRx1 (Sypetkowski et al., 2023) and since unannotated perturbations lack scGPT indices

the authors discard those images, limiting its general applicability, thus the model explores only the annotated perturbations as a factor of variation.

**Methods on the IMPA pipeline.** We refer to the Image Perturbation Autoencoder (IMPA) pipeline as the evaluation/preprocessing setup that applies illumination correction and crops native $512 \times 512$ Cell Painting fields to $96 \times 96$ nuclei-centered patches, with RxRx1 results reported only for U2OS cells. This low-resolution, single–cell-type setting simplifies distribution shift but suppresses organelle-level detail. Within this pipeline, IMPA (Palma et al., 2025) uses an AdaIN-conditional GAN. Although foundational as an early approach, GANs generally exhibit less stable training and lower fidelity than modern diffusion/flow methods (Karras et al., 2022b; Lucic et al., 2018). CellFlux (Zhang et al., 2025) reformulates the task as a distribution-to-distribution mapping from controls to perturbed cells and solves it with conditional flow matching, reporting strong FID/KID results while *following the same IMPA preprocessing* for comparability. Consequently, CellFlux's gains are demonstrated in the same low-capacity crop setting rather than on native-resolution, multi–cell-type fields.

**Comparison to MorphGen.** Prior methods leave critical gaps that MorphGen closes. Unlike MorphoDiff's irreversible RGB compression and IMPA's $96 \times 96$ down-sampling, MorphGen keeps every fluorescence channel intact by wrapping each grayscale slice in a three-channel latent and running diffusion jointly across all six channels. The latents are then split and decoded per channel, preserving organelle detail at the native $512 \times 512$ scale. The resulting higher latent dimensionality is tamed with a representation alignment loss –adapted from REPA (Yu et al., 2025)– but driven by OpenPhenom embeddings (Kraus et al., 2024), instead of generic vision features. This alignment loss stabilizes training and sharpens biological fidelity. Because our perturbation and cell-type embeddings are learned directly from images, MorphGen uses *all* RxRx1 plates (four cell lines, all perturbations), whereas MorphoDiff discards unlabelled perturbations while working only on HUVEC and IMPA is limited to U2OS. Together, full-channel diffusion, alignment loss and data-driven conditioning enable MorphGen to deliver higher-resolution images, generalize across multiple cell types, and reproduce biologically consistent morphologies more faithfully than earlier approaches.

## 3 MORPHGEN

MorphGen is a generative model that synthesizes high-resolution, biologically meaningful cell images under diverse perturbation and cell type conditions. Our model combines a pretrained VAE with a latent diffusion model, adapted to handle multi-channel Cell Painting data. To overcome known issues with high-dimensional latent spaces (e.g. from concatenating six channels) and improve biological fidelity, we introduce an alignment loss inspired by REPA (Yu et al., 2025) and train the diffusion model using features from a microscopy-specific foundation model. This design enables accurate, organelle-resolved synthesis while scaling to large perturbation spaces and multiple cell lines.

We consider a conditional latent diffusion setting for high-resolution, multi-channel fluorescence microscopy images. Let $\mathcal{X} \subset \mathbb{R}^{6 \times H \times W}$ denote the image space of six-channel Cell Painting images, where each channel corresponds to a distinct biological signal.

**Organelle-aware processing.** Since the pretrained VAE encoder is designed for three-channel RGB images, we adapt each grayscale input channel independently by stacking it three times along the channel dimension. Let $\mathbf{x}^{(c)} \in \mathbb{R}^{1 \times H \times W}$ be the $c$-th channel of an image $\mathbf{x} \in \mathcal{X}$. We define its RGB-stacked version as $\tilde{\mathbf{x}}^{(c)} \in \mathbb{R}^{3 \times H \times W}$. This design allows us to encode each organelle-specific channel separately, preserving its biological specificity. Moreover, we retain a latent-space diffusion framework (Rombach et al., 2022) that is easier to train while leveraging powerful pretrained VAEs. By independently encoding each organelle channel while concatenating them into a shared latent for joint modeling, we enable organelle-level generation and analysis that maintain post-generation channel-wise interpretability and capture cross-organelle dependencies.

Each stacked channel $\tilde{\mathbf{x}}^{(c)}$ is passed through a frozen pretrained VAE encoder $E_{\text{VAE}}$ to obtain a compressed latent representation:

$$\mathbf{z}^{(c)} = E_{\text{VAE}}(\tilde{\mathbf{x}}^{(c)}) \in \mathbb{R}^{4 \times H' \times W'}.$$

where $H'$ and $W'$ denote the spatial resolution of the VAE latent space. We then concatenate the six channel-wise latents along the channel dimension to form the full latent representation:

$$\mathbf{z} = \text{concat}(\mathbf{z}^{(1)}, \dots, \mathbf{z}^{(6)}) \in \mathbb{R}^{24 \times H' \times W'}.$$

**Joint diffusion process.** The concatenated latent $\mathbf{z}$, encoding all organelle channels, serves as the input to a latent diffusion model parameterized by a Scalable Interpolant Transformer (SiT) (Ma et al., 2024). Unlike U-Net-based (Ronneberger et al., 2015) architectures traditionally used in diffusion models, SiT operates directly on flattened token sequences and excels at modeling complex spatial relationships through global self-attention. This is particularly advantageous for Cell Painting data, where biological signals are distributed across channels and spatially separated structures. Moreover, transformer-based architectures like SiT offer built-in conditioning mechanisms via class tokens and cross-attention layers, in contrast to the more rigid feature-wise modulation used in U-Nets (Park et al., 2019; Huang & Belongie, 2017). SiT's transformer backbone, combined with its flexible interpolant framework, enables stable training while preserving morphological detail.

Conditioning is achieved through the combination of perturbation, cell type, and diffusion timestep embeddings. Let $p \in \mathcal{P}$ and $ct \in \mathcal{CT}$ denote the perturbation and cell type labels, respectively, which are mapped to learnable embeddings $\mathbf{e}_p, \mathbf{e}_{ct} \in \mathbb{R}^d$. With the timestep embedding $\mathbf{e}_t$, the conditioning vector $\mathbf{c} = \mathbf{e}_p + \mathbf{e}_{ct} + \mathbf{e}_t$ is used as the cross-attention context in SiT. This formulation attributes distinct variations to perturbation and cell-type factors, enabling disentanglement and fine-grained control.

Following the EDM formulation (Karras et al., 2022a), the forward diffusion process generates noisy latent samples by interpolating clean latents $\mathbf{z}_0$ with Gaussian noise. At a randomly sampled timestep $t \in [0, T]$, this interpolation is given by $\mathbf{z}_t = \alpha_t \mathbf{z}_0 + \sigma_t \boldsymbol{\epsilon}$, $\boldsymbol{\epsilon} \sim \mathcal{N}(0, \mathbf{I})$, where $\alpha_t$ and $\sigma_t$ are deterministic scaling factors with boundary conditions $\alpha_0 = \sigma_T = 1$ and $\alpha_T = \sigma_0 = 0$. The model predicts the velocity $\mathbf{v}_t$ of the diffusion trajectory, defined as the time derivative of the latent:

$$\mathbf{v}_t = \frac{d\mathbf{z}_t}{dt} = \dot{\alpha}_t \mathbf{z}_0 + \dot{\sigma}_t \boldsymbol{\epsilon}.$$

Given the noisy latent $\mathbf{z}_t$ and conditioning vector $\mathbf{c}$, SiT $f_\theta$ estimates this velocity: $\hat{\mathbf{v}}_t = f_\theta(\mathbf{z}_t, \mathbf{c})$, and is trained via mean squared error against the ground-truth:

$$\mathcal{L}_{\text{diff}} = \mathbb{E}_{\mathbf{z}_0, t, \boldsymbol{\epsilon}} \left[ \|f_\theta(\mathbf{z}_t, \mathbf{c}) - (\dot{\alpha}_t \mathbf{z}_0 + \dot{\sigma}_t \boldsymbol{\epsilon})\|_2^2 \right].$$

This loss ensures the SiT learns a robust velocity prediction, stabilizing training across noise scales.

**Incorporating biological representations.** To improve semantic consistency and biological fidelity, we adopt representation alignment regularization (REPA) (Yu et al., 2025) during training. Originally, REPA was proposed to align intermediate representations of SiT with strong self-supervised models such as DINOv2 (Oquab et al., 2024), yielding over $17.5\times$ faster convergence. In our case, we adopt the same alignment objective but replace DINOv2 with OpenPhenom (Kraus et al., 2024)–a domain-specific foundation model trained on Cell Painting images. By aligning to OpenPhenom features, we guide the model toward biologically meaningful representations and mitigate the risk of learning spurious patterns unrelated to cellular morphology.

Given an input image $\mathbf{x}$, we extract reference patch-level embeddings: $y^\star = F(\mathbf{x}) \in \mathbb{R}^{N \times d'}$, using OpenPhenom as the feature extractor $F$. Here, $N$ is the number of patches and $d'$ is the embedding dimension. Let $h_t^{(k)} \in \mathbb{R}^{N \times d}$ denote the hidden representation at layer $k$ of the SiT at timestep $t$. This hidden representation is projected through an MLP $h_\phi$ into dimension $d'$ to align with $y^\star$. The REPA loss encourages patch-level alignment via cosine similarity: $\mathcal{L}_{\text{REPA}} = -\frac{1}{N} \sum_{n=1}^{N} \text{sim}\left(y_n^\star, h_\phi(h_{t,n}^{(k)})\right)$.

The total training objective is $\mathcal{L} = \mathcal{L}_{\text{diff}} + \lambda \mathcal{L}_{\text{REPA}}$, where $\lambda$ balances their relative importance. We fix $k = 8$ and $\lambda = 0.5$ in all experiments following Yu et al. (2025).

**Sampling process.** At inference, we generate new six-channel images by running a fixed-step Euler–Maruyama sampler in latent space. Starting from Gaussian noise in concatenated latent space, we step through a sequence of noise levels (by default 50 steps), and at each step use the trained SiT to predict the instantaneous "drift" that moves the latent toward a clean signal. We add a small noise term scaled to the current noise level, then repeat.

Once the final step is reached, we split the $24 \times H' \times W'$ tensor back into six channel-specific latent representations. Each of these is decoded separately through the VAE decoder, yielding six RGB stacks of size $3 \times H \times W$. We then collapse each stack to single grayscale channel by averaging its

three color planes, and recombine all six to form the final $6 \times H \times W$. This simple inverse procedure enables six-channel fluorescent Cell Painting image generation. Importantly, after training, we do *not* use OpenPhenom-based alignment during sampling; generation is independent of OpenPhenom.

**Advantages of the approach.**

- **High-resolution synthesis.** By operating in the latent space of a pretrained VAE combined with powerful latent diffusion models, to the best of our knowledge our method is the first model to jointly synthesize all six Cell Painting channels at resolution $6 \times 512 \times 512$.

- **Organelle-aware generation.** Each fluorescence channel is treated distinctly yet modeled jointly, allowing the model to capture organelle-specific morphology while maintaining spatial and functional coherence across channels.

- **Stable training in high-dimensional latent spaces.** We mitigate the challenges of scaling diffusion to stacked multi-channel latent representations by incorporating an alignment loss guided by a microscopy-specific foundation model's features, improving both stability and generative quality.

- **Flexible compositional conditioning.** Our model supports flexible conditioning on perturbations and cell types simultaneously, enabling controlled generation across biological variables.

## 4 EXPERIMENTS

We design a series of experiments to evaluate the quality, biological fidelity and flexibility of our model. Our evaluations span comparisons with prior state-of-the-art models, generation conditioned on perturbations, cell-types, organelle-specific synthesis, and analysis using image-derived features from CellProfiler (Carpenter et al., 2006) and OpenPhenom (Kraus et al., 2024). We conduct most experiments on RxRx1 (Sypetkowski et al., 2023) –a large scale dataset with perturbation and cell type factors– and to demonstrate cross-dataset generalization, also report results on the Rohban dataset (Rohban et al., 2017) using its 5-gene and 12-gene variants. Detailed descriptions of the datasets and implementation details are provided in the Appendix B and C.

### 4.1 EVALUATION SETUP

**Metrics.** We report Fréchet Inception Distance (FID) and Kernel Inception Distance (KID). Scores are computed using 500 generated vs. 500 real images. All experiments are repeated with three random seeds.

- **Perturbation-level** (Sec. 4.2). To ensure a fair comparison with MorphoDiff, we adopt their experimental protocol. Metrics are independently computed for randomly selected 50 siRNAs; then averaged across perturbations. For RxRx1, although MorphGen natively generates full 6-channel images across multiple cell lines, we restrict it to HUVEC-only generation—matching MorphoDiff's capacity—and convert our outputs to the RGB space using Recursion's visualization script (Sypetkowski et al., 2023).

- **Organelle-specific** (Sec. 4.3). Metrics are computed using the same 50 siRNAs but evaluated in each of the four cell types and in every single channel representing organelles.

- **Cell-type-level** (Sec. 4.3). Metrics are computed per cell type without perturbation conditioning.

**Augmentation policy.** When the real dataset for a given perturbation contained $< 500$ examples, we followed Navidi et al. (2025) and synthetically expanded it using random flips and 90° rotations.

### 4.2 COMPARISON WITH STATE-OF-THE-ART

**Results.** Table 1 demonstrates that, even under MorphoDiff's constrained HUVEC-only, RGB-mapped RxRx1 evaluation, MorphGen achieves substantial improvements: it reduces FID by 64.8 and 27.8, and KID by 0.10 and 0.04, compared to Stable Diffusion and MorphoDiff, respectively. Such large gains under a constrained setting highlight MorphGen's fidelity. Moreover, Figure 6 provides complementary qualitative evidence: across multiple cell-type/perturbation pairs, MorphGen faithfully reproduces the morphology and texture, visually corroborating our quantitative results.

To demonstrate the performance beyond the RxRx1 dataset, we evaluated it on the Rohban dataset under two settings: 5-gene and 12-gene subsets. Similarly, MorphGen achieves state-of-the-art performance in both cases, outperforming prior baselines, as shown in Table 1. On the 5-gene subset, FID

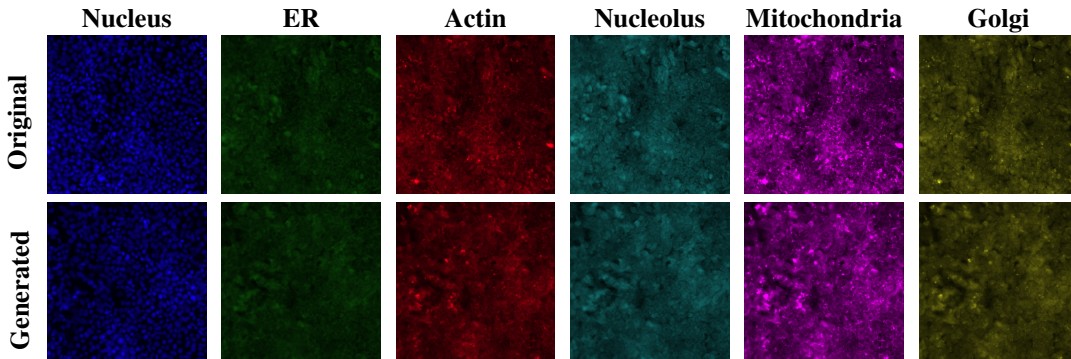

Figure 2: Comparison of original and generated fluorescence images for each organelle in a control HEPG2 cell. Our model reconstructs the six distinct fluorescent channels using RxRx1-recommended colormaps, preserving morphology across subcellular structures. Generated images are not cherry-picked, and we selected original images that are neighbors of the generated ones for visualization.

Table 1: FID and KID (lower is better) across two datasets. RxRx1/HUVEC uses 50 randomly sampled perturbations. Rohban results are reported for the 5-gene and 12-gene subsets.

| Dataset | Subset | Method | FID ↓ | KID ↓ |
|---|---|---|---|---|
| RxRx1 (HUVEC) | 50 perturbations | Stable Diffusion | 115 | 0.11 |
| | | MorphoDiff | 78 | 0.05 |
| | | **MorphGen (Ours)** | $50.2 \pm 2.45$ | $0.01 \pm 0.000$ |
| Rohban dataset | 5 genes | Stable Diffusion | 326 | 0.45 |
| | | MorphoDiff | 251 | 0.33 |
| | | **MorphGen (Ours)** | $100.24 \pm 1.53$ | $0.05 \pm 0.014$ |
| Rohban dataset | 12 genes | Stable Diffusion | 317 | 0.45 |
| | | MorphoDiff | 277 | 0.38 |
| | | **MorphGen (Ours)** | $123.93 \pm 3.51$ | $0.08 \pm 0.017$ |

drops from 251 to $100.24 \pm 1.53$ ($\sim 60\% \downarrow$), on the 12-gene subset, it falls from 277 to $122.93 \pm 3.51$ ($\sim 55\% \downarrow$). Moreover, Appendix G reports the cross-pipeline comparison with low-resolution models.

### 4.3 ADDITIONAL CAPABILITIES

To enable fair comparison across channels and cell types, where target distributions differ, we report a *Relative FID*. This is defined as the ratio between the FID of generated versus real images and the FID of two mutually exclusive real subsets of the same data. A score of 1 indicates generation quality as good as the real distribution. This normalization allows us to interpret performance relative to the inherent variability of each target distribution, rather than relying on incomparable absolute FIDs.

**Organelle-Specific Generation.** To demonstrate MorphGen's fine-grained control, following a similar procedure, we sample 50 siRNAs at random but this time across all four cell types (HEPG2, HUVEC, RPE, U2OS), while matching the original cell-type distribution. For each of the six fluorescence channels (Nucleus, ER, Actin, Cytoplasm, Nucleolus, Mitochondria), we extract real and generated single-channel images, replicate them three times to form 3-channel inputs, and compute FID and KID. As shown in Table 2, the nucleolus channel yields the lowest Relative FID (1.489), successfully modeling the target distribution. In contrast, ER (Relative FID 1.774) is comparatively harder to model. Moreover, Figure 2 provides a qualitative comparison of original versus generated single-channel images in a control HEPG2 cell. The generated outputs closely mirror the morphology and texture of the originals, further demonstrating MorphGen's ability to faithfully reproduce subcellular structures. Appendix L reports the full table with confidence intervals.

**Cell-Type-Specific Generation.** To assess MorphGen under more natural data distributions, we randomly sample images by cell type (HEPG2, HUVEC, RPE, U2OS) without conditioning on perturbations, avoiding any data augmentation to reach a fixed sample count.

Table 2: FID and KID scores for 50 random perturbations across all cell types. Our method supports generation for all four cell types (HEPG2, HUVEC, RPE, U2OS) and provides channel-wise control. MorphoDiff** only supports RGB generation for HUVEC cells.

| Method | Metric | RGB | Nucleus | ER | Actin | Cyto | Nucleolus | Mito |
|---|---|---|---|---|---|---|---|---|
| Ours | FID↓ | **50.2** | 27.6 | 48.1 | 57.6 | 49.6 | 43.6 | 59.0 |
| | KID↓ | **0.008** | 0.010 | 0.011 | 0.015 | 0.013 | 0.012 | 0.012 |
| | Rel. FID↓ | 1.411 | 1.691 | 1.774 | 1.562 | 1.612 | 1.489 | 1.564 |
| MorphoDiff* | FID↓ | 78 | — | — | — | — | — | — |
| | KID↓ | 0.05 | — | — | — | — | — | — |

*MorphoDiff is trained only on the HUVEC cell type and does not support channel-wise generation.

Table 3 reports FID, KID and Relative FID on the resulting RGB-converted images. MorphGen achieves its best scores on HUVEC (Relative FID 1.136) and maintains strong performance across the other cell types (e.g., RPE: Relative FID 1.185). These cell-type-specific results outperform our earlier perturbation-level experiments, which required aggressive augmentation to inflate small-perturbation sets—particularly those with fewer than 50 unique samples—introducing bias into the real data distribution.

Table 3: Per-Cell type specific results. MorphGen is capable of generating high-fidelity images for different cell types.

| Cell Type | FID↓ | KID↓ | Rel. FID↓ |
|---|---|---|---|
| HEPG2 | 41.1 | 0.016 | 1.529 |
| HUVEC | 28.7 | 0.006 | 1.136 |
| RPE | 34.4 | 0.007 | 1.185 |
| U2OS | 38.2 | 0.017 | 1.492 |

By contrast, when following the natural data distribution without artificial augmentation, our model's performance excels further, demonstrating MorphGen's superior fidelity. Please refer to Appendix M for the detailed results with confidence intervals.

## 4.4 Morphology Analysis

**CellProfiler Features.** We evaluated morphological fidelity using CellProfiler (Carpenter et al., 2006), extracting features for real and generated images of the four most common perturbations: 1108, 1124, 1137 and 1138 (control) for HUVEC cells. After variance-thresholding, standardization and removing highly collinear features, we visualize the feature space with PCA and compute correlation matrices over the top-10 PCA-selected features. for both real and generated sets.

**Biological Validity.** Figure 3 shows, generated samples closely align with the real distribution within each perturbation, indicating MorphGen preserves perturbation-specific morphology. In addition, the pairwise correlation structure of key CellProfiler features closely matches between real and generated data shown in Figure 4. This suggests MorphGen captures biologically meaningful relationships rather than only marginal statistics. See Appendices I and P for analyses on downstream classification performance (quantitative) and additional visualizations (qualitative).

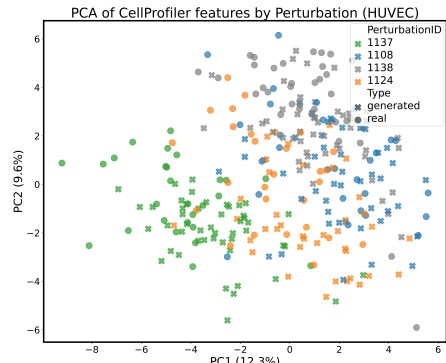

Figure 3: **PCA of CellProfiler features.** Color denotes perturbation (1108, 1124, 1137, 1138); marker style denotes data type (circle: real, cross: generated). Generated samples align with real clusters while maintaining perturbation separation.

**OpenPhenom Features.** To further assess the biological plausibility of our generated images, we extend our analyses using the microscopy-specific foundation model OpenPhenom. We focus on the same four perturbations in the dataset (1108, 1124, 1137 and 1138) and visualize both real and generated samples in a shared PCA embedding space. The resulting visualizations are shown in Figure 5. Similarly, we observe that (i) real and generated embeddings largely overlap within the same perturbation class, indicating that generated images reproduce morphology faithfully, and (ii) perturbation-specific deviations from the control are similarly captured in both real and generated distributions,

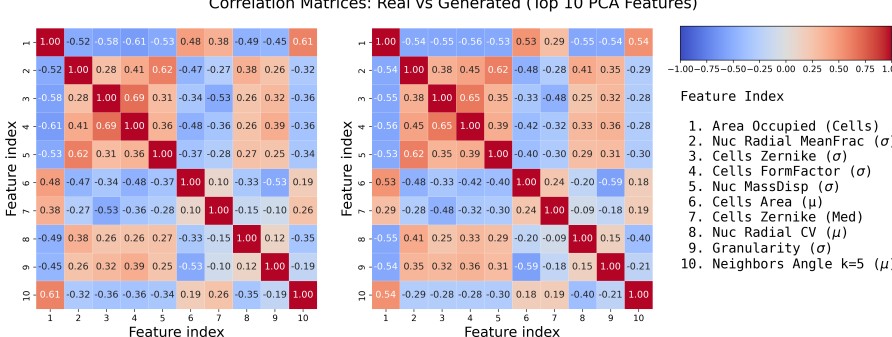

**Figure 4: CellProfiler morphology analysis (HUVEC).** Correlation matrices for the top-10 PCA-selected features in real and generated data, shown side-by-side with a shared scale, indicate that MorphGen preserves key morphological relationships.

evident from the clear color separation. Note that during inference, images are generated without OpenPhenom alignment and processed independently. Moreover, the consistency between Figures 3 and 5 further indicates that OpenPhenom representations are well aligned with CellProfiler features, which was also claimed in the original paper. These patterns suggest that our generative model successfully encodes biologically relevant phenotypic variation while preserving class-level consistency. See Appendices F and I for additional analyses on feature variance and downstream performance.

**CATE: Average Treatment Effect in Feature Space.** To quantitatively validate that our generated images capture biologically meaningful perturbation effects (Bereket & Karaletsos, 2024) at the population level, we compute the Conditional Average Treatment Effect (CATE) between control and perturbed samples using OpenPhenom features. We use OpenPhenom features to represent cellular morphology following Kraus et al. (2024) and denote the image-level embedding as $Y$, obtained by averaging patch-level representations across the image. Given a perturbation $p$, we define CATE as associational difference between a treated population and a control group for a specific cell type:

$$\text{CATE}(p) = \left\| \mathbb{E}[Y \mid P = \text{control}, \, ct = \text{HUVEC}] - \mathbb{E}[Y \mid P = p, \, ct = \text{HUVEC}] \right\|^2$$

This metric captures the squared Euclidean distance between the average feature vectors of the control group (perturbation 1138) and a perturbed group $p$. We compute the CATE separately for real and generated samples across the three most common perturbations: 1108, 1124, and 1137. While clearly the image-level embeddings do not correspond directly to biological quantities where the treatment effect can immediately be interpreted, Kraus et al. (2024) showed that the OpenPhenom features are very strong predictors of the CellProfiler (Carpenter et al., 2006) features.

Therefore, we estimate the Average Treatment Effect in feature space, as the associational difference will carry over to any downstream prediction-powered (morphological) predictor (Cadei et al., 2024; 2025). We remark that the goal of this experiment is to show that the conclusions drawn from population-level statistics of morphological features match between real and generated samples. Specific causal conclusions may still be invalid, depending on whether hidden confounding or consistency, exchangeability, and overlap assumptions hold on the real data.

Table 4: Conditional Average Treatment Effect (CATE) between control (p1138) and perturbed samples, computed using real and generated HUVEC images.

| Comparison | $\text{CATE}_{\text{real}}$ | $\text{CATE}_{\text{gen}}$ | $\Delta\text{CATE}\downarrow$ |
|---|---|---|---|
| p1138 vs p1137 | 7.85 | 7.41 | 0.43 |
| p1138 vs p1124 | 2.13 | 2.31 | 0.18 |
| p1138 vs p1108 | 0.44 | 0.38 | 0.06 |

As shown in Table 4, perturbation 1137 results in the largest morphological deviation from the control, while perturbation 1108 has the smallest effect. These magnitudes align well with the spatial patterns observed in the PCA projections (Figures 3 and 5). The close agreement between CATE values computed from real and generated images further indicates that our model reliably captures biologically meaningful perturbation effects. This experiment, conclusively shows that they do, and that conclusions from statistical associations at the population level between real and generated images match well. For organelle and cell-type specific visualizations and CATE results, refer to Appendices O and N.

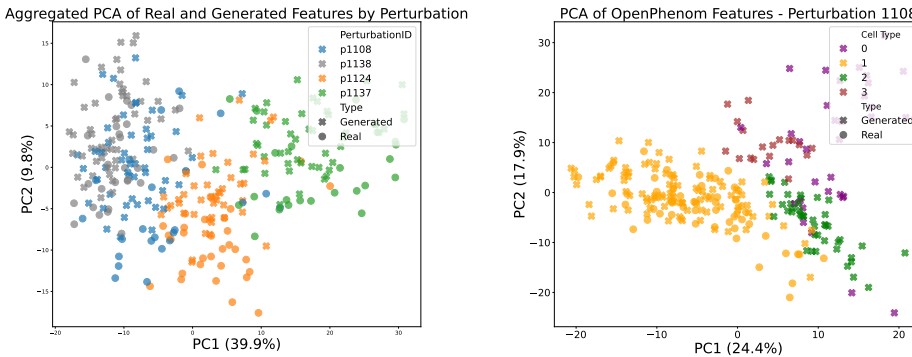

Figure 5: PCA projections of OpenPhenom features from real and generated images. The left panel shows the joint distribution of most frequent perturbations (including the control, p1138) for HUVEC cells, with points colored by perturbation. The right panel visualizes the perturbation 1108 across different cell types. In both panels, marker shapes indicate whether the sample is real or generated.

## 4.5 VIRTUAL INSTRUMENT

We repeat the model training and morphological analysis, leaving out perturbation 1137 on HUVEC from the training set. We selected this particular combination as this is the most frequent cell type in the dataset and, of the four most frequent perturbations, the one with the largest CATE. In other words, MorphGen has seen many images of this cell type (albeit without this perturbation), and this perturbation, which has a strong effect, was applied to many other samples from other cell types. Perhaps unsurprisingly, we observe no significant performance drop on the held-out group, with FID (38.14 vs. 38.07) and $\Delta$CATE (0.46 vs. 0.43) remaining nearly unchanged.

Our model naturally benefits from its much more diverse training set, meaning that this compositional generalization problem is actually relatively close to the training distribution. While this is not a conclusive experiment to test the validity of MorphGen as a virtual instrument, it is an encouraging proof-of-concept, showing that scaling generative models on diverse data has the potential to generalize to unseen experiments, which is a prerequisite for serving as virtual instruments. In Appendix K we provide the detailed setup and FID/KID baselines showing that MorphGen remains consistent with the real distribution even without exposure to this combination during training.

## 5 CONCLUSION

We introduced MorphGen, a generative model for synthesizing high-resolution, six-channel Cell Painting images that preserve biologically meaningful structure across diverse perturbations and cell types. Our model is trained at scale on the full RxRx1 dataset and incorporates a novel alignment loss that leverages embeddings from a microscopy-specific foundation model (OpenPhenom). This alignment guides the diffusion process toward more biologically faithful image synthesis.

MorphGen advances prior work by addressing both image quality and generalization. Quantitatively, it achieves significantly improved FID and KID scores compared to existing models, while also covering a broader generative space defined by two key biological factors: perturbation and cell type. Beyond visual fidelity, features extracted from MorphGen-generated images exhibit population-level trends consistent with real data. In particular, we show that conditional average treatment effects (CATEs) computed from synthetic images align closely with those derived from real ones, supporting their utility in downstream phenotypic analyses.

A missing result is the generalization to uncommon perturbations, however, beyond HUVEC, RxRx1 lacks sufficient samples for reliable evaluation, so larger datasets are needed. Although larger datasets exist (e.g. RxRx3 (Fay et al., 2023)), they cover only one cell type. Even then, MorphGen also does not yet extrapolate to entirely novel conditions. As future work, we will explore instance-based conditioning with learning conditioning embeddings directly from image examples to remove the need of perturbation labels. Despite these limitations, MorphGen represents a meaningful step toward the virtual instruments, accelerating experimental design in functional genomics and drug discovery.

## 6 ETHICS STATEMENT

All datasets used in this work (Sypetkowski et al., 2023; Rohban et al., 2017) are publicly available and were commonly used and processed by prior works (Palma et al., 2025; Rohban et al., 2017; Sypetkowski et al., 2023; Zhang et al., 2025; Navidi et al., 2025). While the generated samples from MorphGen are partially validated using Cell Profiler and OpenPhenom, the samples, especially for combinations of cells and perturbations not in the training data should not be interpreted as definitive biological ground truth without further experimental validation. To mitigate potential misuse, we emphasize that the work is intended for advancing computational methodology in machine learning, not for direct clinical application.

## 7 REPRODUCIBILITY STATEMENT

We provide an anonymized code repository in the Supplementary Materials (full training/evaluation scripts, configs, and environment files) to exactly regenerate all tables and figures. Datasets and preprocessing are detailed in Appendix B. Full implementation details (SD-VAE preprocessing and scaling constants, SiT-XL/2 backbone, REPA-style alignment to OpenPhenom), optimization/sampling settings, precision, EMA, and hardware are in Appendix C (with architectural specifics under Architectural Details). All experiments are repeated with three random seeds: 0, 7 and 1337.

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

## A   TRAINING AND INFERENCE PIPELINE

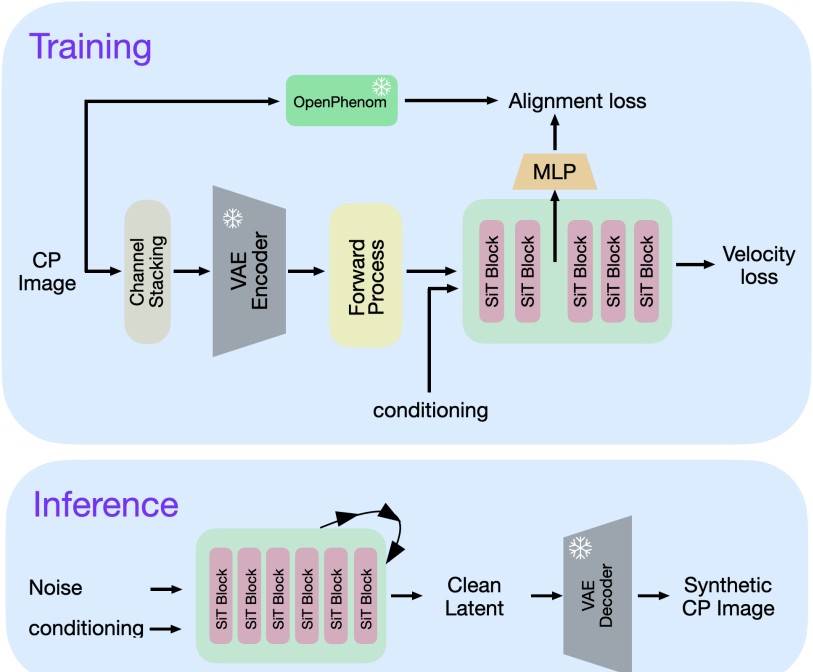

Figure 6: **Overview of MorphGen. Top (training).** A Cell Painting (CP) image is first processed with channel stacking and encoded by a frozen VAE encoder into a multi-channel latent. The latent is perturbed by the diffusion forward process and fed, together with timestep, cell type, and perturbation embeddings (conditioning), into a SiT-based backbone. An MLP projects the SiT features to the OpenPhenom space to compute the alignment loss, while the diffusion head is trained with a velocity loss on the noisy latent. **Bottom (inference).** Starting from noise and the same conditioning embeddings, the SiT backbone iteratively denoises the latent to a clean latent, which is decoded by the frozen VAE decoder to obtain a synthetic CP image.

## B   DATASET

### B.1   RxRx1 DATASET.

This dataset is a large-scale, high-resolution collection of fluorescence microscopy images designed to support the study of phenotypic cellular responses to gene knockdowns and to benchmark batch effect correction methods (Sypetkowski et al., 2023). It comprises $125,510$ images from four human cell types (HUVEC, RPE, HepG2 and U2OS), each exposed to one of $1,108$ siRNA treatments targeting distinct genes, along with 30 non-targeting control conditions. Imaging is performed using a modified Cell Painting assay, generating six-channel $512 \times 512$ pixel images that visualize major subcellular structures including the nucleus, endoplasmic reticulum, actin cytoskeleton, nucleoli, mitochondria and golgi apparatus. By capturing morphological changes induced by gene-specific knockdowns, RxRx1 serves as a challenging benchmark for models aiming to generalize across perturbations, cell types and experimental batches.

### B.2   ROHBAN DATASET.

This dataset comprises Cell Painting images of U2OS cells after ORF overexpression of 323 genes. Following MorphoDiff, we adopted the same preprocessing protocol (including uniform resizing of $512 \times 512$). For analyses, we focus on two small gene subsets: (i) 5 genes from pathways reported to

affect cellular morphology, and (ii) 12 genes selected via clustering in Rohban et al. (2017) based on morphological features.

This dataset comprises Cell Painting images of U2OS cells after ORF overexpression of 323 genes. Following MorphoDiff (Navidi et al., 2025), we adopted the same preprocessing protocol (including uniform resizing of $512 \times 512$). For analyses, we focus on two small gene subsets: (i) 5 genes from pathways reported to affect cellular morphology (2250 images), and (ii) 12 genes selected via clustering in Rohban et al. (2017) based on morphological features (3150 images).

## C   IMPLEMENTATION DETAILS

MorphGen is trained as a latent-diffusion model operating on SD-VAE Rombach et al. (2022) latents of six-channel RxRx1 (Sypetkowski et al., 2023) training set images of $512 \times 512$ resolution. Each single-channel grayscale image is stacked into a 3-channel RGB format, scaled to $[-1, 1]$, encoded with the public stabilityai/sd-vae-ft-mse VAE (AI, 2022), and rescaled by the SD constants $(0.18215, 0)$. The diffusion backbone is a Scalable Interpolant Transformer (SiT XL/2) (Ma et al., 2024). During training, OpenPhenom (Kraus et al., 2024) embeddings of the raw image are injected via a REPA-style projection loss (weight = 0.5), mirroring the original REPA formulation (Yu et al., 2025).

Optimization uses AdamW ($\beta = 0.9/0.999$, lr $= 1 \times 10^{-4}$) (Loshchilov & Hutter, 2018) in mixed precision (fp16 or bf16) under HuggingFace Accelerate (Gugger et al., 2022); TF32 kernels can be enabled for additional speed. An EMA (Karras et al., 2024) shadow network (decay = 0.9999) is maintained for sampling. Sampling uses a 50-step Euler–Maruyama schedule (Ma et al., 2024).

Training runs with a batch size of 16 for up to 400 k steps using 8 H100 GPUs. Checkpoints are written every 50 k steps, and the best model is chosen by the average FID across distributed workers, computed on 100 real vs. generated images of pre-selected perturbations using Inception-V3 (Szegedy et al., 2016) features. Logging and image grids are tracked in Weights and Biases (Biewald, 2020), and all hyper-parameters, metrics, and checkpoints are stored for full reproducibility.

### C.1   ARCHITECTURAL DETAILS

**KL-regularized VAE.** The KL-regularized VAE used in *stabilityai/sd-vae-ft-mse* is a fully convolutional encoder-decoder that compresses a $3 \times 512 \times 512$ RGB image into a $4 \times 64 \times 64$ latent tensor and then reconstructs it. The encoder begins with a $3 \times 3$ convolution followed by three residual blocks, each ending in a stride 2 down-convolution, so channels scale $128 \rightarrow 256 \rightarrow 512$ while the spatial size shrinks $512 \rightarrow 256 \rightarrow 128 \rightarrow 64$. A $1 \times 1$ convolution converts the 512-channel map into eight channels holding per-pixel mean and log-variance; sampling 4-channel latent from these statistics is implemented through the reparameterization trick. The decoder mirrors the encoder: a $1 \times 1$ post-quant convolution restores 512 channels, and a final $3 \times 3$ convolution followed by tanh maps to RGB. The network contains  81 million learnable parameters ( 42M in the encoder,  39M in the decoder).

**OpenPhenom.** OpenPhenom Kraus et al. (2024) is a channel-agnostic masked autoencoder (CA-MAE) built on a *Vision Transformer Small (ViT-S* backbone with $16 \times 16$ patching. It tokenizes each image into patch embeddings, and passes them through 12 transformer layers (with hidden size 384 and 6 heads). A channel-wise cross-attention block lets information flow between stains, and after masking most patches during pre-training the lightweight decoder learns to reconstruct them. At inference time the decoder is dropped and the encoder outputs a fixed 384-dimensional embedding for either one vector per image, or one per channel for finer control. The entire network contains around 22 millions parameters, making it compact for downstream transfer learning.

**Scalable Interpolant Transformer (SiT).** SiT is a diffusion/flow-hybrid generator that swaps the U-Net of classic DDPMs for a Vision-Transformer backbone. It first splits a latent image into $2 \times 2$ patches, yielding a sequence of tokens that pass through 28 transformer blocks with 1152-dimensional hidden size and 16-head self-attention. Each block uses adaptive layernorm-zero modulation: the sum of a sinusoidal timestep embedding and a learned class label embedding is projected and applied as per-channel shifts/scales/gates, letting the same weights serve any diffusion step or class. The encoder is followed by a small MLP projectors which outputs intermediate features for multi-scale

predictions, while a final layer unpatchifies the sequence back to an image-shaped residual that guides the interpolant sampler. SiT-XL retains 675 million parameters, yet consistently attains lower FID thanks to the more flexible interpolant objective and an Euler-Maruyama sampler tuned post-training.

## D  Use of Large Language Models

We used LLMs only for language and LaTeX assistance (editing, phrasing and small utility snippets). All scientific content; methods, experiments, analyses, and results was produced, verified and is reproducible by the authors without LLMs.

## E  Ablations

**Channel-wise Processing.** To evaluate how different preprocessing strategies affect reconstruction fidelity using a pretrained VAE, we compared three settings. In the baseline setup (RGB), all six fluorescence channels are first compressed into an RGB image before encoding, as done in prior work like MorphoDiff. In contrast, our organelle-aware strategy (MorphGen) processes each channel independently by repli-

Table 5: Comparison of reconstruction fidelity across different channel processing strategies using a frozen VAE.

| Channel Processing | MSE $\downarrow$ |
|---|---|
| RGB | $7.13 \times 10^{-4}$ |
| Organelle-aware | $4.93 \times 10^{-5}$ |
| Organelle-aware + RGB | $4.06 \times 10^{-4}$ |

cating it across RGB channels to match the VAE's input expectations. We then consider two ways of reporting reconstruction loss. First, we compute the per-channel reconstruction loss and average across channels, yielding a remarkably low MSE of $4.93 \times 10^{-5}$. Second, we recompose the reconstructed channels into a 6-channel image and apply the same RGB conversion used in the baseline, resulting in an MSE of $4.06 \times 10^{-4}$. As shown in Table 5, both variants outperform the RGB baseline, demonstrating that organelle-aware processing enables better use of the pretrained VAE and leads to consistently improved fidelity—even under the same evaluation transformation.

**Morphgen without the alignment loss.** Table 6 presents an ablation study evaluating the effect of incorporating an alignment loss on Open-Phenom features. While Yu et al. (2025) introduced a similar alignment strategy to accelerate diffusion training, we instead leverage it to guide the model toward learning biologically meaningful representations during generation. With the alignment loss, MorphGen achieves more than a 10% reduction in FID and over a 20% reduction in KID, indicating a substantial improvement in image fidelity. These results support the effectiveness of our proposed approach in integrating biological priors through pretrained features during training.

Table 6: Ablation study on the alignment loss. We compare models trained without (top row) and with (bottom row) alignment regularization. FID and KID scores (lower is better) are reported for 50 randomly sampled perturbations from the HUVEC cell type.

| Method | FID$\downarrow$ | KID$\downarrow$ |
|---|---|---|
| MorphGen wo/ align. | $56.87 \pm 3.35$ | $0.023 \pm 0.001$ |
| MorphGen | $\mathbf{50.2 \pm 2.45}$ | $\mathbf{0.018 \pm 0.000}$ |

## F    GENERATION VARIANCE EXPERIMENT

The qualitative analysis done on the feature representations of the generated vs real images (Figure 5) shows the spread on the principal components which shows meaningful variance across conditioned classes. Additionally, we conducted an feature variance analysis (on HUVEC, perturbations 1108, 1124, 1137 and 1138) where we extracted the features using OpenPhenom for both real and generated samples and measured their variances to check if the feature variances distribute similarly to the real data.

Table 7: Feature variance analysis comparing features extracted by real and generated images.

| Treatment | Real Variance | Gen Variance | Difference |
|---|---|---|---|
| p1108 | 0.0171 | 0.0144 | $-0.0027$ |
| p1124 | 0.0145 | 0.0155 | $+0.0010$ |
| p1137 | 0.0173 | 0.0192 | $+0.0019$ |
| p1138 | 0.0129 | 0.0159 | $+0.0030$ |

**Takeaway.** It can be seen from the Table 7 the generated variance is similar to (or even slightly higher) than the one of real data, indicating that our model preserves diversity and does not exhibit mode collapse.

## G    COMPARISON WITH CELLFLUX

**CellFlux (Zhang et al., 2025)**   introduces a generative framework that treats cellular morphology prediction as a distribution-to-distribution mapping from same-batch control images to perturbed cells, solved via conditional flow matching. Evaluated on BBBC021 (chemical), RxRx1 (genetic), and JUMP (combined) datasets, CellFlux reports substantially lower FID/KID following IMPA's evaluation pipeline.

Despite its strong results, like IMPA, it focuses on low-resolution ($6 \times 96 \times 96$) crops and a single cell type (U2OS), whereas MorphGen synthesizes native-resolution $6 \times 512 \times 512$ fields across all four RxRx1 cell types. To make CellFlux comparable with MorphGen, we adopt two strategies:

1. **Relative FID**. We report the ratio $\mathrm{FID}(\mathrm{gen}, \mathrm{real})/\mathrm{FID}(\mathrm{real}_A, \mathrm{real}_B)$, which normalizes target-distribution shifts due to IMPA's preprocessing and differing resolutions (lower is better; values near 1 indicate parity).

2. **IMPA-matched evaluation**. Ee extract $96 \times 96$ nuclei-centered crops from MorphGen outputs using Otsu thresholding and compare against IMPA's preprocessed real images. This deliberately conservative protocol disadvantages MorphGen, which was trained on full-resolution, channel-normalized images with different illumination statistics.

Table 8: Relative FID comparison across *different* target distributions. MorphGen is evaluated on 96×96 crops from native, channel-normalized data; CellFlux uses IMPA's pipeline (U2OS, 96×96).

| Method | FID (gen–real) | FID (real–real) | Relative FID |
|--------|----------------|-----------------|--------------|
| MorphGen | $205.470 \pm 2.242$ | $144.743 \pm 4.608$ | $1.417 \pm 0.052$ |
| CellFlux | $168.797 \pm 0.868$ | $80.712 \pm 0.444$ | $2.091 \pm 0.021$ |

## G.1 RELATIVE FID COMPARISON

Using Relative FID within the same target pipeline, we normalize away preprocessing and resolution effects. Table 8 shows that MorphGen is significantly closer to its own target distribution than CellFlux is to IMPA's ($1.417 \pm 0.052$ vs. $2.091 \pm 0.021$. Because absolute FIDs are tied to the chosen target distribution and preprocessing pipeline, they are not directly comparable across methods; Relative FID corrects for this dependency and thus provides a fair basis for cross-pipeline comparison.

## G.2 IMPA-MATCHED EVALUATION

Table 9: CellFlux vs. MorphGen under IMPA-matched evaluation. Means $\pm$ 95% CI over $n$=3 seeds. Lower is better.

| Method | FID (gen–real) | FID (real–real) | Relative FID |
|--------|----------------|-----------------|--------------|
| CellFlux | $168.797 \pm 0.868$ | $80.712 \pm 0.444$ | $2.091 \pm 0.021$ |
| MorphGen | $182.970 \pm 0.464$ | $80.660 \pm 0.716$ | $2.260 \pm 0.025$ |

Even under the disadvantaged analysis (IMPA-style 96×96 U2OS crops) that disadvantages MOR-PHGEN—trained on native-resolution 6×512×512 channel-normalized images—the methods are close. The real–real baselines are effectively identical (80.712±0.444 vs. 80.660±0.716), confirming matched targets, and MORPHGEN's Relative FID is within single-digit percent of CellFlux (2.260 vs. 2.091). This suggests MORPHGEN remains competitive outside its native resolution/preprocessing regime while additionally supporting multi–cell-type, full-resolution synthesis.

**Takeaway.** Using Relative FID, we enable a fair cross-pipeline comparison. Results show that even when focusing only on the zoomed $96 \times 96$ crops of our higher-resolution images, MorphGen is significantly closer to its own target distribution—indicating sensitivity to fine-grained detail at native resolution. Furthermore, when evaluated against a different target distribution using the IMPA pipeline for full comparability, MorphGen performs on par with CellFlux despite this deliberately disadvantaged setting.

## H PERFORMANCE IN LOW-SAMPLE REGIMES

To complement our analysis on the frequently observed perturbations (p1108, p1124, p1137, p1138), we also evaluate MorphGen's performance in a low-sample regime. We report the class-conditional FIDs with only 16 samples per perturbation in Table 10.

Table 10: Class-conditional FID with 16 samples per set.

| Perturbation ID | Num. Samples | FID (real–real) | FID (gen–real) |
|-----------------|--------------|-----------------|----------------|
| 789 | 16 | 105.55 | 112.65 |
| 862 | 16 | 105.39 | 120.66 |
| 83 | 16 | 121.65 | 127.53 |
| 531 | 16 | 104.33 | 109.25 |
| 1048 | 16 | 119.61 | 112.55 |

*Note:* With only 16 images per set, class-conditional FID can fluctuate by ±10–15 points, so some gen–real scores may appear marginally lower (e.g., perturbation 1048) than the corresponding real–real baselines.

**Takeaway.** Results show that FIDs between generated and real sets closely match those computed between independent, mutually exclusive real subsets, highlighting MorphGen's generation quality even under data scarcity.

## I   ANALYSIS OF DOWNSTREAM PERFORMANCE ON THE CONDITIONING FACTORS

Beyond generative quality, MorphGen can populate balanced datasets for downstream tasks such as classification, clustering, and causal discovery. To assess the downstream utility and the biologically meaningful distinctions captured by our generator, we conducted the following experiments.

### I.1   OPENPHENOM FEATURES

We extracted 384-dimensional OpenPhenom embeddings from both real Cell Painting images and MorphGen-generated images using the frozen OpenPhenom encoder. A simple logistic regression probe was then trained under three train $\rightarrow$ test regimes:

- **Real $\rightarrow$ Real**: baseline setting where both train and test data are real.
- **Generated $\rightarrow$ Generated**: both train and test sets are synthetic MorphGen images.
- **Generated $\rightarrow$ Real**: cross-domain setting where the probe is trained on synthetic images and evaluated on held-out real images.

We evaluated the probe on two prediction tasks—cell type (4 classes) and perturbation ID (4 classes)— reporting test accuracy as well as 5-fold cross-validation (CV) mean and standard deviation on the training split.

Table 11: Linear probe experiment with OpenPhenom features on the most frequent 4 perturbations: 1108, 1124, 1137 and 1138.

| Train $\rightarrow$ Test | Task | Test Acc. | CV Mean | CV Std | # Train | # Test |
|---|---|---|---|---|---|---|
| Real $\rightarrow$ Real | Cell-type | 0.997 | 0.992 | 0.0049 | 760 | 327 |
| | Perturbation | 0.838 | 0.845 | 0.0245 | 760 | 327 |
| Generated $\rightarrow$ Generated | Cell-type | 0.994 | 0.999 | 0.0024 | 840 | 360 |
| | Perturbation | 0.858 | 0.838 | 0.0208 | 840 | 360 |
| Generated $\rightarrow$ Real | Cell-type | 0.983 | 0.998 | 0.0020 | 1200 | 1087 |
| | Perturbation | 0.804 | 0.848 | 0.0308 | 1200 | 1087 |

**Takeaway.** OpenPhenom features enable near-perfect cell-type classification and robust perturbation recognition even when the probe is trained solely on MorphGen images and evaluated on real data, indicating that the generator preserves biologically relevant signal.

### I.2   CELLPROFILER FEATURES

We extracted morphological features for HUVEC cells for the four most frequent perturbations (1108, 1124, 1137, 1138) using a standard CellProfiler pipeline (modules such as IdentifyPrimary/Secondary/TertiaryObjects, MeasureGranularity, MeasureObjectIntensity/Neighbors/SizeShape, MeasureTexture, MeasureImageAreaOccupied and MeasureImageIntensity). Starting from the full feature set (2533 features), we applied the following preprocessing: (i) z-score standardization, (ii) variance thresholding at 1e-5 to remove near-constant features and (iii) correlation filtering at $|r| \geq 0.7$ to eliminate redundancy. This reduced the feature space to 878 and then 70 features, respectively. We then applied PCA and kept 32 components.

Similar to the OpenPhenom setting, we trained a linear classifier to predict perturbation ID (4 classes) under the same three regimes. We report the accuracy and macro-F1 scores in Table 12.

Table 12: Downstream perturbation classification with CellProfiler features (HUVEC, 4 perturbations: 1108, 1124, 1137 and 1138)

| Train → Test | Accuracy | Macro-F1 |
|---|---|---|
| Real → Real | 0.794 | 0.792 |
| Generated → Generated | 0.873 | 0.873 |
| Generated → Real | 0.739 | 0.734 |

It can be seen in Table 12 that Generated→Generated yields the strongest scores (accuracy $\sim$ 0.87), confirming that MorphGen produces internally consistent samples with clear perturbation separability. In contrast, Real→Real shows lower scores ($\sim 0.79$) and visibly higher variability, which is expected given natural measurement noise and biological heterogeneity in real data. Importantly, Generated→Real performs *close to* Real→Real (0.74 vs. 0.79), indicating that classifiers trained on generated images recover perturbation-discriminative structure that aligns well with the real measurements.

**Takeaway.** Overall, this supports MorphGen's success at capturing biologically meaningful, perturbation-specific signal rather than mere visual plausibility.

## J   BENEFIT OF SYNTHETIC DATA FOR DOWNSTREAM CLASSIFICATION

To further examine the utility of MorphGen for downstream tasks, we tested whether augmenting training sets with synthetic images can improve perturbation classification performance. We extended the probe from the 4 most frequent perturbations to the 10 frequent ones, thereby increasing phenotypic similarity between classes and making the classification problem more challenging. This setting provides a more demanding benchmark for evaluating the benefit of synthetic augmentation.

### J.1   OPENPHENOM FEATURES

We trained a logistic regression probe on OpenPhenom embeddings under four regimes:

- **Gen → Gen**: both train and test sets consist of MorphGen images (sanity check).
- **Gen → Real**: train only on synthetic images, test on real images.
- **Real → Real**: baseline with real images only.
- **Real+Gen → Real**: same as Real → Real, but with the training set augmented by synthetic images.

Table 13: Linear probe accuracy on the 10 frequent perturbations using OpenPhenom features.

| Perturbation | Gen → Gen | Gen → Real | Real+Gen → Real | Real → Real |
|---|---|---|---|---|
| 1108 | 0.633 | 0.607 | 0.706 | 0.706 |
| 1109 | 0.813 | 0.821 | 0.813 | 0.813 |
| 1110 | 0.800 | 0.375 | 0.647 | 0.588 |
| 1111 | 1.000 | 0.750 | 0.882 | 0.824 |
| 1112 | 0.808 | 0.250 | 0.588 | 0.529 |
| 1113 | 0.849 | 0.964 | 0.824 | 0.765 |
| 1114 | 0.800 | 0.429 | 0.647 | 0.588 |
| 1115 | 0.813 | 0.536 | 0.492 | 0.529 |
| 1119 | 0.735 | 0.196 | 0.625 | 0.625 |
| 1138 | 0.758 | 0.470 | 0.900 | 0.800 |
| **Average** | 0.801 | 0.540 | 0.712 | 0.677 |

In this harder 10-class setting, the Real → Real baseline drops from 83.8% to 67.7%. Augmenting the training set with MorphGen images increases accuracy by +3.6 percentage points (67.7% → 71.2%).

Six out of ten classes improve, three remain unchanged, and one decreases slightly. Training exclusively on synthetic images still achieves 54% accuracy, showing that MorphGen-generated data carry discriminative morphological signal despite the domain gap.

**Takeaway.** Synthetic augmentation with MorphGen improves real-data classification accuracy, while training on generated images alone still yields features that generalize well to real images.

## J.2    RESNET18 FEATURES

Although OpenPhenom embeddings are used only during training for alignment, evaluating downstream performance on the same representation space can raise concerns about circularity. To address this, we repeated the probe experiment using a ResNet18 feature extractor trained independently of MorphGen.

Table 14: Linear probe accuracy on the 10 frequent perturbations using ResNet18 features.

| Perturbation ID | Real → Real | Gen → Real | Real+Gen → Real |
|---|---|---|---|
| 1108 | 0.529 | 0.353 | 0.353 |
| 1109 | 0.563 | 0.563 | 0.500 |
| 1110 | 0.294 | 0.353 | 0.353 |
| 1111 | 0.529 | 0.765 | 0.706 |
| 1112 | 0.471 | 0.235 | 0.529 |
| 1113 | 0.412 | 0.412 | 0.529 |
| 1114 | 0.588 | 0.294 | 0.529 |
| 1115 | 0.294 | 0.412 | 0.412 |
| 1119 | 0.375 | 0.250 | 0.375 |
| 1138 | 0.500 | 0.600 | 0.700 |
| **Average** | **0.456** | **0.424** | **0.499** |

When evaluated on ResNet18 features, the Real → Real baseline reaches 45.6% accuracy, and augmenting with synthetic images improves it to 49.9%. Training exclusively on generated data achieves 42.4%. Despite overall lower absolute accuracies compared to OpenPhenom (reflecting weaker features), the relative trends remain consistent: synthetic augmentation provides a measurable gain, and generated data alone still capture class-discriminative signal.

**Takeaway.** This confirms that the observed benefits are not an artifact of the OpenPhenom feature space.

## K    VIRTUAL INSTRUMENT EXPERIMENT DETAILS

Virtual Instrument experiment evaluates MorphGen's ability to generalize to an unseen (cell type, perturbation) combination. Specifically, perturbation 1137 on HUVEC was removed from the training set, while all other data remained available. The resulting FID values compare identical target images but two different models:

- **38.07** – full-data model: generated (HUVEC, p1137) vs. real (HUVEC, p1137),

- **38.14** – held-out-combination model: generated (HUVEC, p1137) vs. the same real set.

To contextualize these numbers, on Table 15 we show FID and KID values between 250-image subsets of real data under self and cross-perturbation comparisons.

The FID of approximately 38 for the virtual instrument setting lies much closer to the self-comparison baseline (20.9) than to any cross-perturbation regime. This indicates that the generated distribution remains consistent with the held-out real distribution, despite the absence of this combination from training. While this analysis is not a comprehensive evaluation of MorphGen as a virtual instrument, it provides quantitative evidence that the model exhibits compositional generalization in this setting.

Table 15: Cross-perturbation FIDs and KIDs between real images (250 samples per set).

| Comparison | FID | KID Mean | KID Std |
|---|---|---|---|
| p1137 vs. p1138 | 262.19 | 0.3733 | 0.0192 |
| p1137 vs. p1137 (self) | 20.89 | 0.0003 | 0.0009 |
| p1137 vs. p1108 | 165.91 | 0.1778 | 0.0126 |
| p1137 vs. p1124 | 100.49 | 0.0722 | 0.0063 |

**Takeaway.** The near-identical FID of the held-out model and the full-data model, and its closeness to the self-FID baseline rather than cross-perturbation values, shows that MorphGen generalizes well to the unseen (cell type, perturbation) combination.

## L ORGANELLE SPECIFIC FULL RESULTS

Table 16 reports the same metrics as Table 2, with the addition of 95% confidence intervals.

Table 16: FID and KID scores (mean $\pm$ 95% CI) for 50 random perturbations across all cell types. Our method supports generation for all four cell types (HEPG2, HUVEC, RPE, U2OS) and provides channel-wise control.

| Channel | FID $\downarrow$ | KID $\downarrow$ | Rel. FID $\downarrow$ |
|---|---|---|---|
| RGB | $50.2 \pm 1.3$ | $0.0082 \pm 0.0003$ | $1.411 \pm 0.085$ |
| Nucleus | $27.6 \pm 0.6$ | $0.0101 \pm 0.0008$ | $1.691 \pm 0.206$ |
| ER | $48.1 \pm 0.4$ | $0.0116 \pm 0.0008$ | $1.774 \pm 0.128$ |
| Actin | $57.6 \pm 1.2$ | $0.0155 \pm 0.0003$ | $1.562 \pm 0.032$ |
| Cyto | $49.6 \pm 0.4$ | $0.0132 \pm 0.0005$ | $1.612 \pm 0.092$ |
| Nucleolus | $43.6 \pm 0.4$ | $0.0123 \pm 0.0009$ | $1.489 \pm 0.099$ |
| Mito | $59.0 \pm 2.3$ | $0.0121 \pm 0.0018$ | $1.564 \pm 0.155$ |

## M CELL-TYPE SPECIFIC FULL RESULTS

Table 17 reports the same metrics as Table 3 with the addition of 95% confidence intervals.

Table 17: FID, KID, and Relative FID scores (mean $\pm$ 95% CI) across cell types. MorphGen generates high-fidelity images consistently across diverse cell types.

| Cell Type | FID $\downarrow$ | KID $\downarrow$ | Rel. FID $\downarrow$ |
|---|---|---|---|
| HEPG2 | $41.11 \pm 2.46$ | $0.016 \pm 0.0005$ | $1.529 \pm 0.055$ |
| HUVEC | $28.65 \pm 1.98$ | $0.006 \pm 0.0003$ | $1.136 \pm 0.086$ |
| RPE | $34.35 \pm 1.67$ | $0.007 \pm 0.0003$ | $1.185 \pm 0.059$ |
| U2OS | $38.17 \pm 2.27$ | $0.017 \pm 0.0004$ | $1.492 \pm 0.204$ |

## N CELL-TYPE-SPECIFIC CATE AND VISUALIZATIONS WITH OPENPHENOM FEATURES

Table 18 shows the Conditional Average Treatment Effect (CATE) between control (p1138) and perturbed samples, computed using real and generated images across different cell types using OpenPhenom features. Results indicate that images generated by MorphGen preserve treatment-specific cellular features with high fidelity, closely mirroring those from real images. Unsurprisingly, the consistency is strongest for HUVEC cells, likely due to their higher representation in the dataset. The highest treatment effect (i.e., deviation from p1138) in real samples is observed for HUVEC under treatment p1137, with a CATE of 7.85. MorphGen-generated images closely match this effect with a

Table 18: Conditional Average Treatment Effect (CATE) between control (1138) and perturbed samples, reported per cell type.

| Cell Type | p1138 vs p1137 | | | p1138 vs p1108 | | | p1138 vs p1124 | | |
|---|---|---|---|---|---|---|---|---|---|
| | $CATE_{real}$ | $CATE_{gen}$ | $\Delta CATE$ | $CATE_{real}$ | $CATE_{gen}$ | $\Delta CATE$ | $CATE_{real}$ | $CATE_{gen}$ | $\Delta CATE$ |
| HEPG2 | 1.07 | 1.06 | 0.01 | 1.19 | 0.48 | 0.71 | 1.27 | 0.98 | 0.29 |
| HUVEC | 7.85 | 7.41 | 0.44 | 0.44 | 0.38 | 0.06 | 2.13 | 2.31 | 0.18 |
| RPE | 1.28 | 1.09 | 0.19 | 1.00 | 0.65 | 0.35 | 1.04 | 0.83 | 0.21 |
| U2OS | 3.53 | 2.77 | 0.76 | 0.38 | 0.34 | 0.04 | 3.46 | 2.42 | 1.04 |

CATE of 7.41, demonstrating consistency even in cases of strong perturbation response. Overall, the closeness of real and generated CATE values suggests that MorphGen-generated images can support accurate downstream analysis.

REAL VS GENERATED

Figure 7 presents PCA visualizations of OpenPhenom Kraus et al. (2024) features for four representative perturbations – 1108, 1124, 1137 and the control 1138 – arranged in rows. The left column compares real and generated samples by coloring the points by image type. The strong overlap between real and generated distributions across all perturbations indicates that the generated images faithfully reproduce the morphological feature space of the real data. Whereas the right column colors the same embeddings by cell type, revealing clear separability across cell types. This suggest that OpenPhenom features, even when derived from generated images, retain meaningful cell-type structure and can support downstream analyses such as classification. Together, these results demonstrate that MorphGen produces high-quality, morphologically accurate samples that preserve both perturbation effects and intrinsic cell-type differences.

Figures 8, 9, 10, and 11 show PCA visualizations at the cell-type level (HEPG2, HUVEC, RPE, and U2OS, respectively), allowing a qualitative assessment of color separation across different perturbations. Overall, the results demonstrate that features extracted from MorphGen-generated images exhibit: (i) strong alignment with real features, making them visually indistinguishable, and (ii) clear separability with respect to both cell type and perturbation—the two generative factors we explicitly control.

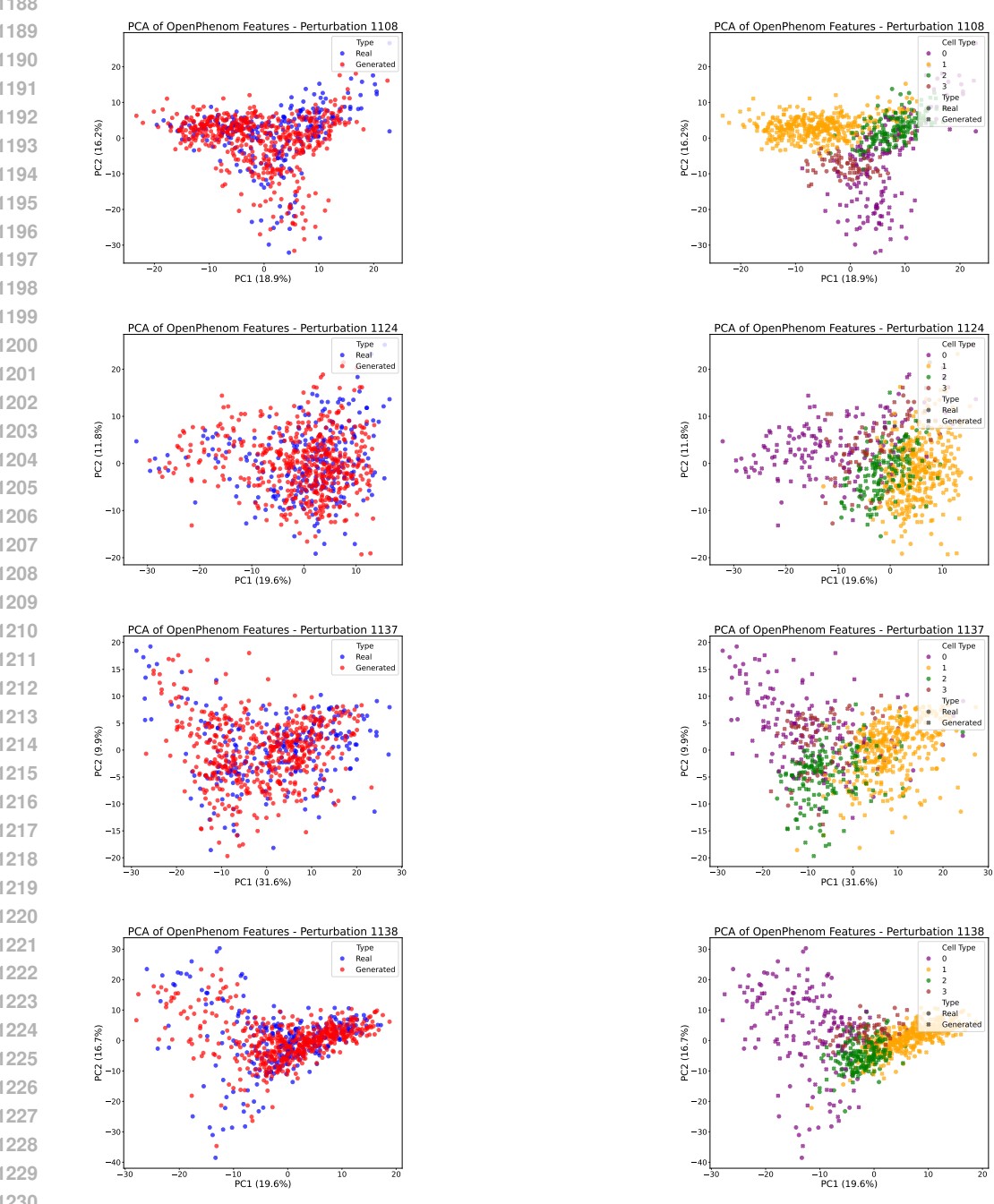

Figure 7: PCA visualizations of OpenPhenom features for four perturbations (rows: 1108, 1124, 1137, and control 1138). Each row compares real and generated embeddings for a single perturbation. *Left column:* points are colored by image type (real vs. generated), revealing strong overlap—indicating that generated images closely match the distribution of real samples. *Right column:* the same embeddings are now colored by cell type, showing that cell-type-specific structure is preserved in both real and generated data. Together, these plots demonstrate that our model produces high-quality, morphologically faithful samples that capture both perturbation effects and intrinsic cell type differences.

PERTURBATION EFFECTS VISUALIZATIONS HEPG2

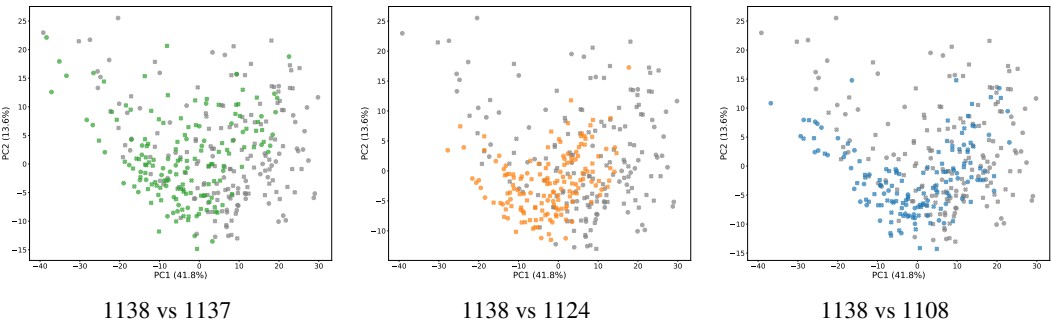

(a) PCA visualization across all four perturbations, including the control (1138).

1138 vs 1137    1138 vs 1124    1138 vs 1108

Figure 8: PCA projections of phenotypic embeddings of HEPG2 cells. The top panel shows global variation across all perturbations. The bottom panels show focused pairwise comparisons between the control (1138) and specific perturbations.

PERTURBATION EFFECTS VISUALIZATONS HUVEC

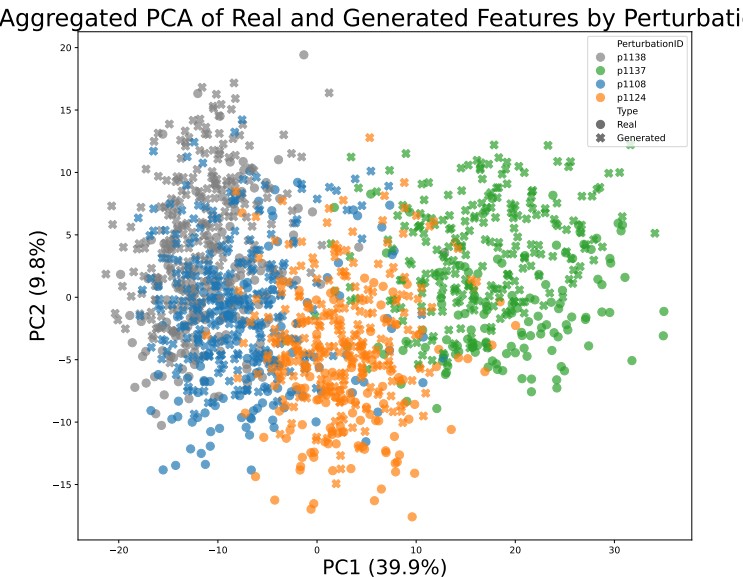

(a) PCA visualization across all four perturbations, including the control (1138).

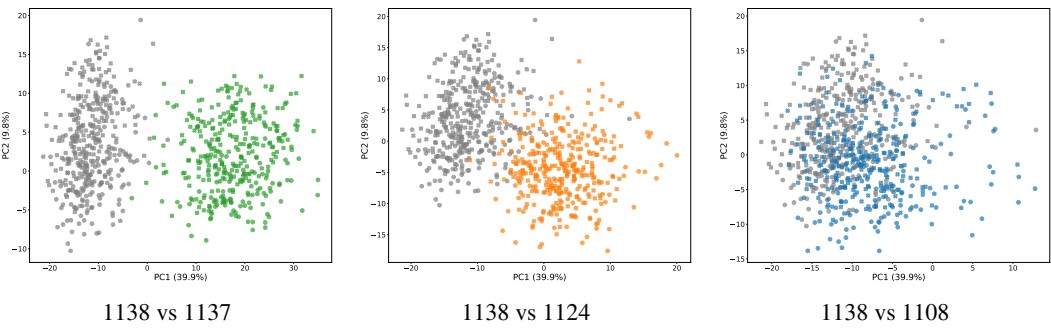

1138 vs 1137          1138 vs 1124          1138 vs 1108

Figure 9: PCA projections of phenotypic embeddings of HUVEC cells. The top panel shows global variation across all perturbations. The bottom panels show focused pairwise comparisons between the control (1138) and specific perturbations.

PERTURBATION EFFECTS VISUALIZATIONS RPE

### Aggregated PCA of Real and Generated Features by Perturbation

(a) PCA visualization across all four perturbations, including the control (1138).

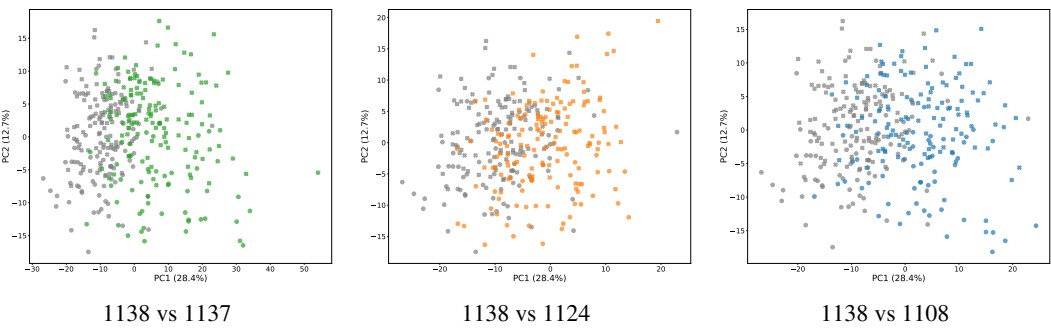

1138 vs 1137        1138 vs 1124        1138 vs 1108

Figure 10: PCA projections of phenotypic embeddings of RPE cells. The top panel shows global variation across all perturbations. The bottom panels show focused pairwise comparisons between the control (1138) and specific perturbations.

PERTURBATION EFFECTS VISUALIZATIONS U2OS

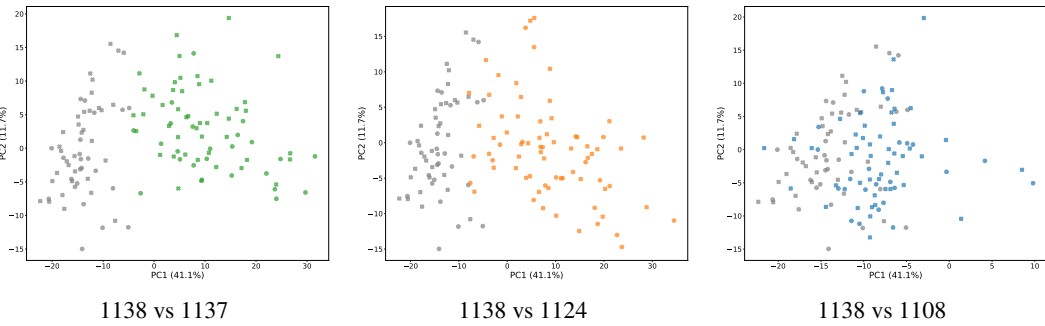

(a) PCA visualization across all four perturbations, including the control (1138).

1138 vs 1137          1138 vs 1124          1138 vs 1108

Figure 11: PCA projections of phenotypic embeddings of U2OS cells. The top panel shows global variation across all perturbations. The bottom panels show focused pairwise comparisons between the control (1138) and specific perturbations.

# O  ORGANELLE-SPECIFIC CATE AND VISUALIZATIONS WITH OPENPHENOM FEATURES

Table 19 reports organelle-specific CATEs, measuring the deviation of perturbations p1108, p1124, and p1137 from the control p1138. Notably, our estimates closely match the real values even at the organelle level, suggesting that MorphGen can accurately capture organelle-specific response patterns. Figure 12 further illustrates this by showing PCA visualizations of Nuclei responses across different perturbations.

Table 19: Conditional Average Treatment Effect (CATE) between control (1138) and perturbed HUVEC cells, reported per organelle.

| Organelle | p1138 vs p1137 | | | p1138 vs p1108 | | | p1138 vs p1124 | | |
|---|---|---|---|---|---|---|---|---|---|
| | $\text{CATE}_{\text{real}}$ | $\text{CATE}_{\text{gen}}$ | $\Delta\text{CATE}$ | $\text{CATE}_{\text{real}}$ | $\text{CATE}_{\text{gen}}$ | $\Delta\text{CATE}$ | $\text{CATE}_{\text{real}}$ | $\text{CATE}_{\text{gen}}$ | $\Delta\text{CATE}$ |
| Nuclei | 9.47 | 9.50 | 0.03 | 0.69 | 0.59 | 0.10 | 2.86 | 2.97 | 0.11 |
| ER | 9.46 | 9.54 | 0.08 | 0.69 | 0.60 | 0.09 | 2.85 | 2.97 | 0.12 |
| Actin | 9.49 | 9.50 | 0.01 | 0.69 | 0.59 | 0.10 | 2.86 | 2.97 | 0.11 |
| Nucleoli | 9.49 | 9.51 | 0.02 | 0.69 | 0.59 | 0.10 | 2.85 | 2.96 | 0.11 |
| Mitochandria | 9.54 | 9.51 | 0.03 | 0.70 | 0.59 | 0.11 | 2.86 | 2.97 | 0.11 |
| Golgi | 9.49 | 9.52 | 0.03 | 0.69 | 0.60 | 0.09 | 2.86 | 2.97 | 0.11 |

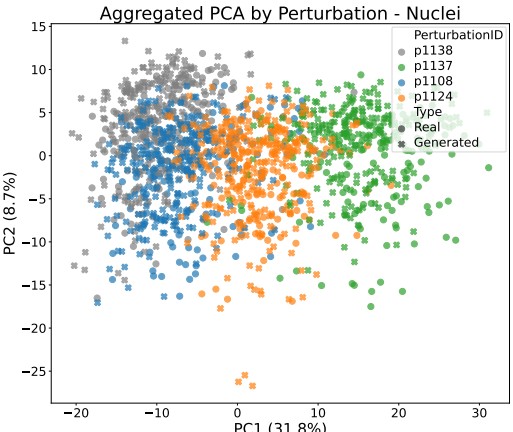

(a) PCA visualization across all four perturbations, including the control (1138).

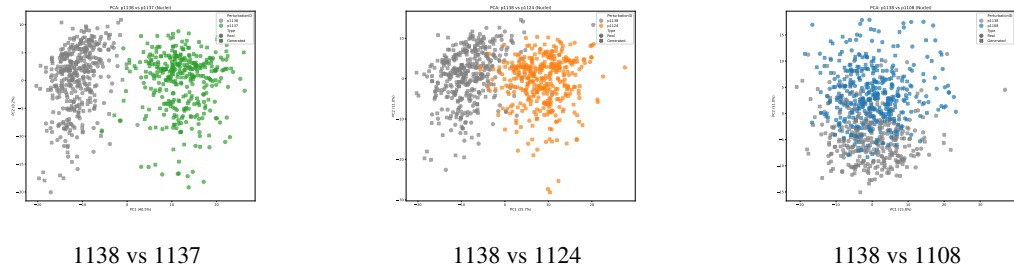

1138 vs 1137          1138 vs 1124          1138 vs 1108

Figure 12: PCA projections of Nuclei embeddings of HUVEC cells. The top panel shows global variation across all perturbations. The bottom panels show focused pairwise comparisons between the control (1138) and specific perturbations.

# P    PCA VISUALIZATIONS WITH CELLPROFILER FEATURES

To further validate the fidelity of generated morphologies, we present PCA visualizations of CellProfiler features for four representative perturbations. Figure 13 shows strong alignment between real and generated samples, highlighting MorphGen's fidelity.

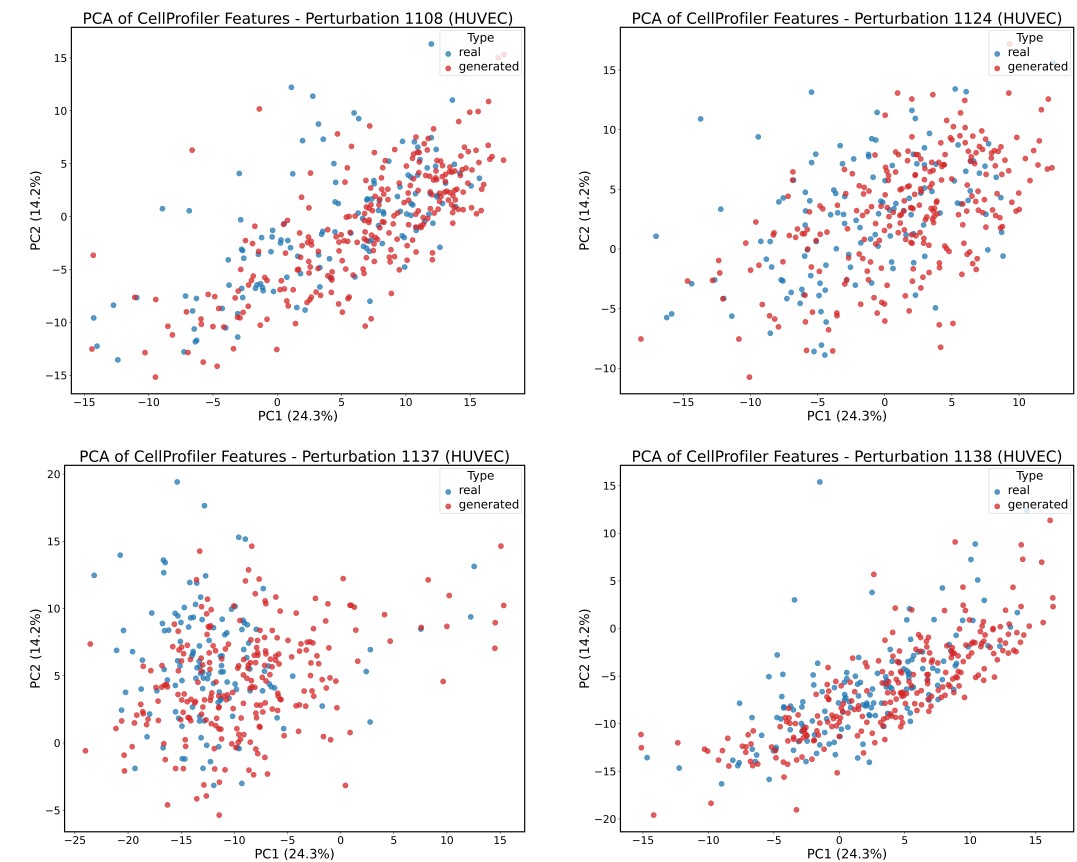

Figure 13: **PCA of CellProfiler features for selected perturbations (HUVEC).** Colors indicate real (blue) and generated (red) samples, the strong overlap highlights that generated images closely match the distribution of real samples.

# Q QUALITATIVE EXAMPLES

In this section, we report additional qualitative evidence of the quality of samples generated with MorphGen. Figures 14, 15, 16 and 17 present representative examples across multiple perturbations and cell types, illustrating that MorphGen consistently produces realistic and diverse cellular morphologies.

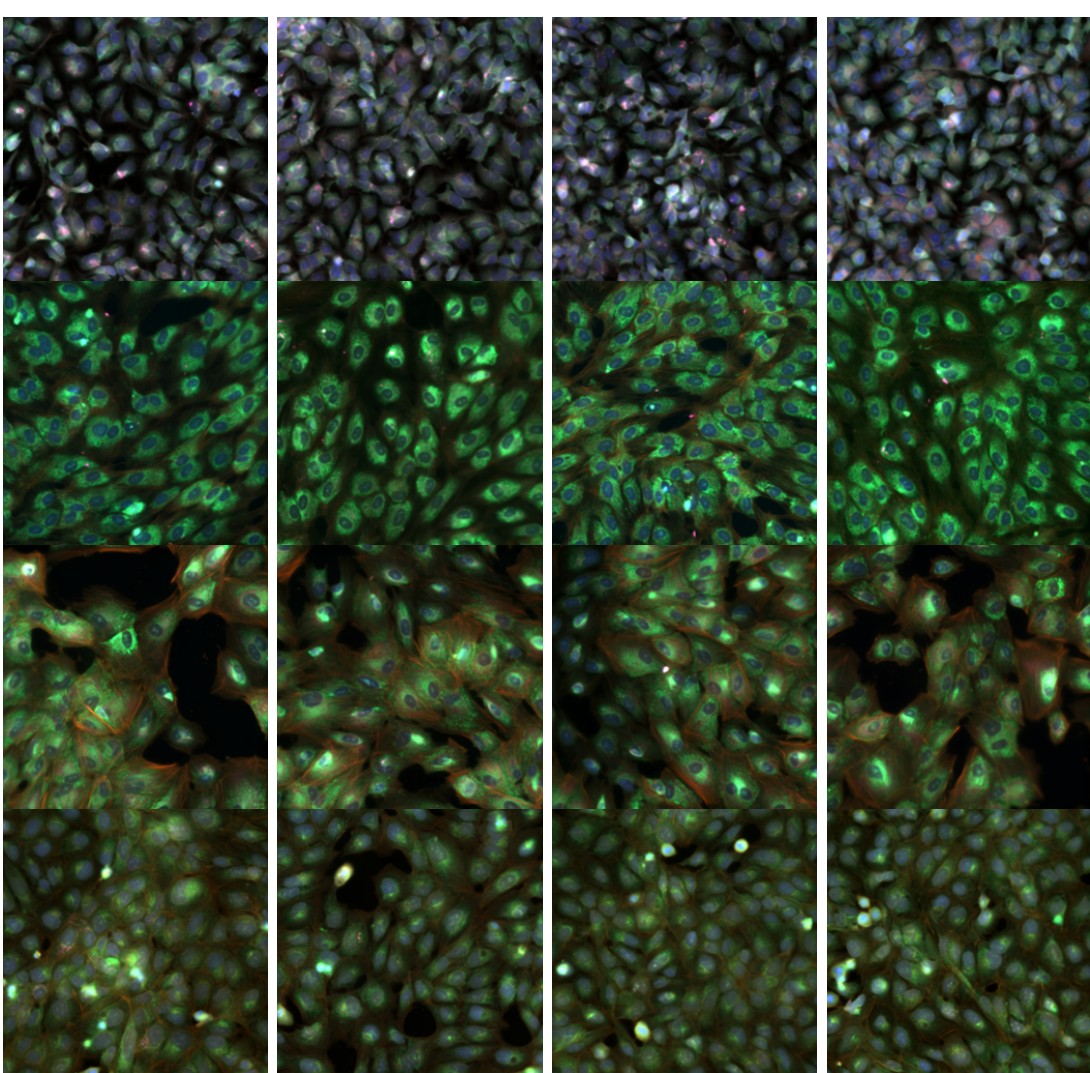

Figure 14: Qualitative examples of generated cell images for perturbation **1108**. Rows correspond to different cell types (HEPG2, HUVEC, RPE, U2OS), and columns show four independent samples per cell type.

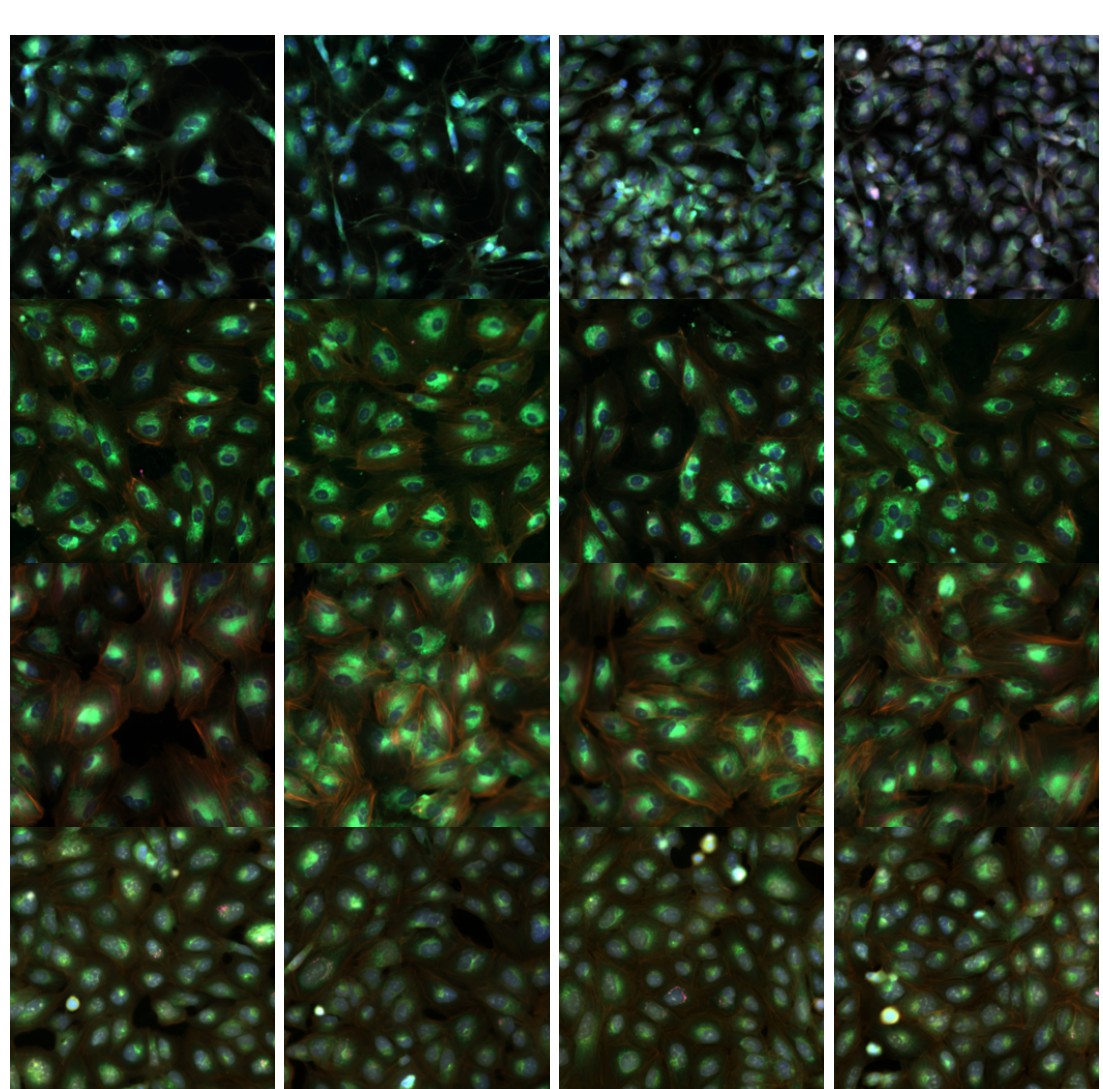

Figure 15: Qualitative examples of generated cell images for perturbation **1124**. Rows correspond to different cell types (HEPG2, HUVEC, RPE, U2OS), and columns show four independent samples per cell type.

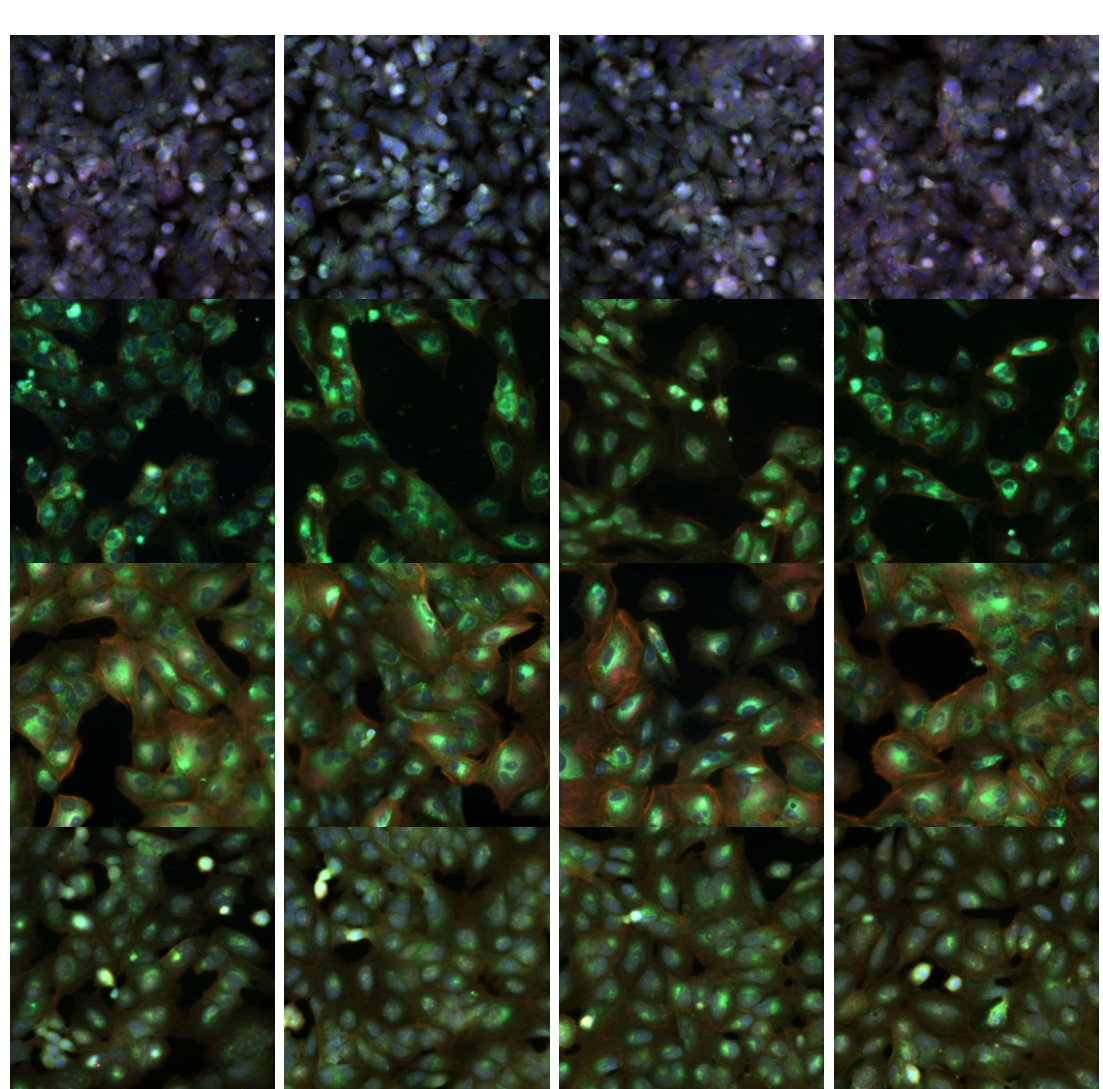

Figure 16: Qualitative examples of generated cell images for perturbation **1137**. Rows correspond to different cell types (HEPG2, HUVEC, RPE, U2OS), and columns show four independent samples per cell type.

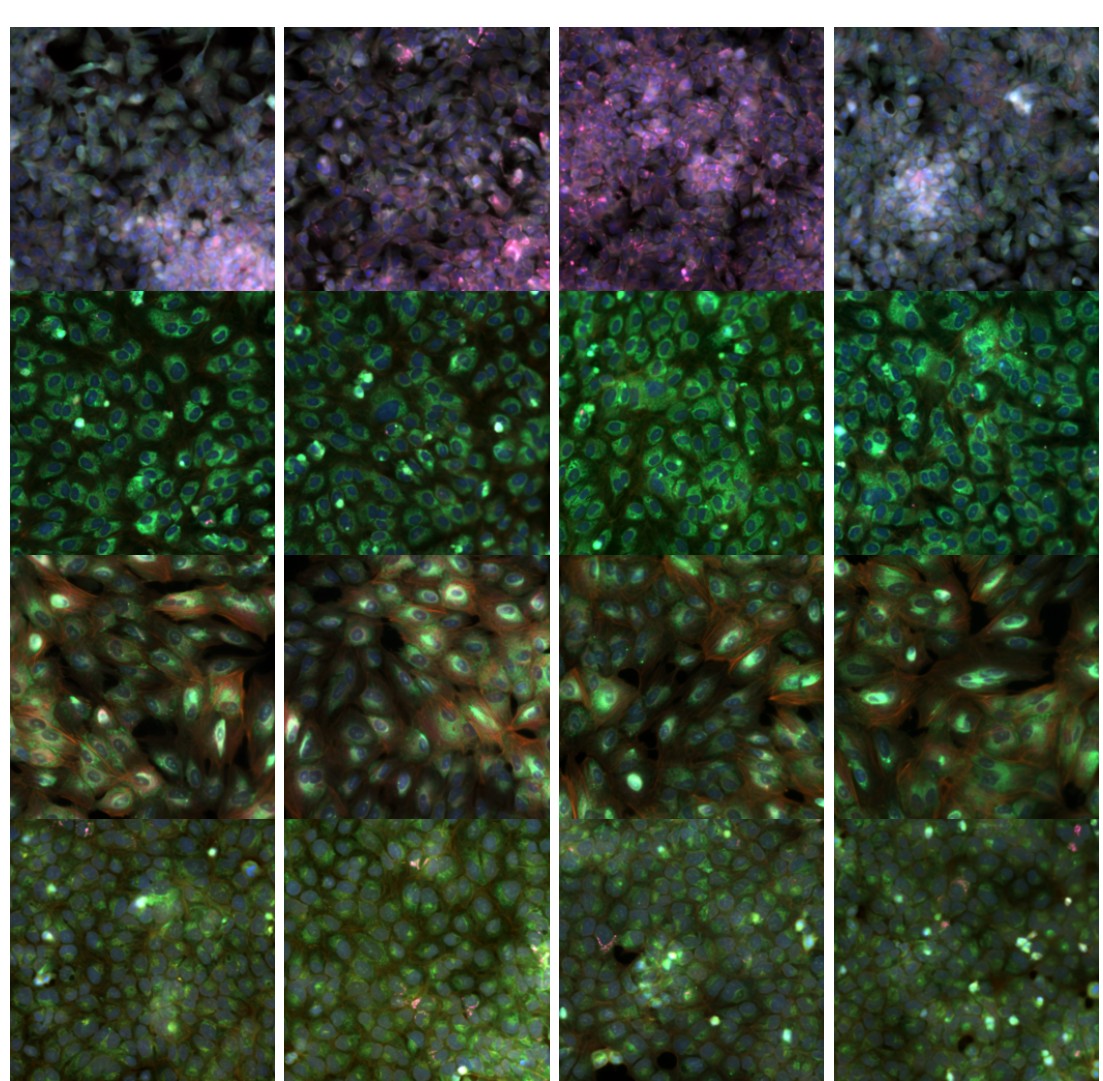

Figure 17: Qualitative examples of generated cell images for perturbation **1138**. Rows correspond to different cell types (HEPG2, HUVEC, RPE, U2OS), and columns show four independent samples per cell type.

