# OpenReview forum: "MorphGen: Controllable and Morphologically Plausible Generative Cell-Imaging"
_ICLR.cc/2026/Conference — Submitted to ICLR 2026_

### Official Review · Reviewer_g5zt · 2025-10-30

**Soundness:** 2
**Presentation:** 3
**Contribution:** 2
**Rating:** 2
**Confidence:** 4

**Summary:**

In this work, the authors propose Morphegen, a generative model to predict the morphological cellular responses to perturbations. They introduce strategies to make this framework compatible with the multi-channel nature of HCS imaging. The authors performed a series of experiments to validate the performance of the method.

**Strengths:**

1. This work tackles an interesting problem, namely the application of generative models to HCA images. Indeed, these types of images generally exceed the three channels found in standard RGB images, which requires adapting general-purpose generative models to this kind of data.

2. The paper is well written and easy to follow.

3. The authors present interesting ideas for adapting  latent diffusion models to HCA images.

**Weaknesses:**

1. While I find the method interesting, the novelty appears limited, as it mainly consists of adapting Morphodiff to biological images with more than three channels.

2. The related works section lacks several important methods that address the prediction of cellular responses to perturbations [1,2].

3. The proposed baseline is somewhat weak, as the authors only compare their model to Morphodiff and Stable Diffusion, reporting FID and KID scores. Moreover, these metrics are related: FID is typically suited for large datasets, whereas KID is more appropriate when working with fewer images.

4. The evaluation is based on only two datasets, which may limit the robustness of the conclusions.

5. The authors do not provide any schematic to describe the proposed architecture, and such a schematic would greatly facilitate understanding.

References:

[1] PhenDiff: Revealing Subtle Phenotypes with Diffusion Models in Real Images, Bourou et al.

[2] Revealing invisible cell phenotypes with conditional generative modeling, Lamiable et al.

**Questions:**

1. It is unclear what the authors mean when they state: “Our model combines a pretrained VAE with a latent diffusion model.” A latent diffusion model already includes a VAE that encodes the image into a lower-dimensional latent representation, where the diffusion process is performed. Do the authors refer to this built-in VAE, or are they introducing an additional one? Furthermore, on which images was the VAE pretrained, and how many channels were used during pretraining?

2. Previous methods [1,2] were already able to generate biologically meaningful images. What improvement does REPA provide in this case? Did the authors perform an ablation study to evaluate the importance of each component?

3. How is the conditioning performed? Which encoders are used to encode the different perturbations?

4. The U-Net used in modern diffusion models already includes self-attention blocks to capture spatial relationships. Does SiT provide any improvement over this? Did the authors compare the two backbones?

5. FID and KID were originally proposed to evaluate RGB image generation. How do the authors apply these metrics to images with more than three channels?

6. I do not understand how Stable Diffusion was used as a baseline, since it does not include any encoder to handle perturbation conditioning. How is this achieved in the proposed setup? Furthermore, Stable Diffusion is trained on natural RGB images, so it seems unreasonable to apply it directly to biological images without retraining. Did the authors fine-tune or adapt the model in any way? What about the text encoder?

7. In Table 1, was the number of channels the same for all methods? Although MorphenGen provides the best FID, the values remain very high. Furthermore, why did the authors not provide standard deviations for the other models?

References:

[1] PhenDiff: Revealing Subtle Phenotypes with Diffusion Models in Real Images, Bourou et al.

[2] Revealing invisible cell phenotypes with conditional generative modeling, Lamiable et al.

---

> ### Author Response · Authors · 2025-11-19
>
> We thank the reviewer for the thoughtful and constructive comments. Below, we address the key concerns and outline the revisions made to strengthen the manuscript.
>
> > While I find the method interesting, the novelty appears limited, as it mainly consists of adapting Morphodiff to biological images with more than three channels.
>
> We believe incorporating **domain features into otherwise generic latent spaces** using a foundation model through representation alignment to improve the generation quality is a novel idea. We used REPA loss to achieve this alignment to inject the correct semantic features to guide the diffusion process.
>
> This is particularly useful as in Cell Painting images, we have high-resolution images with non native (not RGB) channels, as the increase in latent dimensionality would impede the diffusion performance. To the best of our knowledge, MorphGen is the first model that achieves (i) native 6-channel 512x512 generation and (ii) multi-factor conditioning (cell type + perturbation). Ablations in Table 6 show that full-channel generation + alignment significantly improves FID/KID and, most importantly, biological faithfulness (CellProfiler / CATE) over RGB/unaligned variants.
>
>
> > The related works section lacks several important methods that address the prediction of cellular responses to perturbations [1,2].
>
> We will add a discussion about image to *image translation* models in our related work, including [1] and [2]. However, we clarify that MorphGen is *purely a generative model* that does not require a control sample to generate an image, and therefore it is not directly comparable with style transfer models.
>
> > The proposed baseline is somewhat weak, as the authors only compare their model to Morphodiff and Stable Diffusion, reporting FID and KID scores. Moreover, these metrics are related: FID is typically suited for large datasets, whereas KID is more appropriate when working with fewer images.
>
> We believe this demonstrates the strength of the approach. We have comparisons including MorphoDiff [5], CellFlux [4], IMPA [3], and Stable Diffusion (from [5]) across 2 datasets and 3 different setups.
>
> Moreover, MorphGen, amongst the others, is the most general one achieving high resolution synthesis on multiple cell types, while retaining state-of-the-art performance. The evaluation setup we compared against (MorphoDiff) is from the ICLR 2024 Spotlight. Therefore, we disagree that the baselines are weak.
>
> Following the prior work [5], we matched the evaluation when possible and reported FID, KID and Relative FID to provide comparability with other methods. Moreover, we included in depth analysis with stress tests on downstream classification task performance, biological faithfulness using CellProfiler features, CATE consistency, as well as many PCA plots (distinguishable clusters w.r.t. cell type and perturbation, while indistinguishable clusters of real vs generated samples) where all agree on generation quality of MorphGen.
>
> > The evaluation is based on only two datasets, which may limit the robustness of the conclusions.
>
> As listed in the conclusion section, the availability of datasets with multi-cell-type + perturbation factors is limited and RxRx1 is the benchmark we found that offers this combination at large scale. We additionally demonstrated results on Rohban variants (5 and 12 gene experiments) for a smaller scale validity. Compared to the literature, using 2 datasets is not particularly narrow. Specifically, if you consider that many methods discard HEPG2, RPE and U2OS cell lines from RxRx1, to the best of our knowledge, our paper provides the largest scale results in the literature, in terms of sample size and data diversity (multiple cell lines and all perturbations in RxRx1).
>
> > The authors do not provide any schematic to describe the proposed architecture, and such a schematic would greatly facilitate understanding.
>
> Thank you for the suggestion, you can now see the schematic (Fig. 6) in the updated PDF.

---

> > ### Author Response · Authors · 2025-11-19
> >
> > Questions:
> >
> > > It is unclear what the authors mean when they state: “Our model combines a pretrained VAE with a latent diffusion model.” A latent diffusion model already includes a VAE that encodes the image into a lower-dimensional latent representation, where the diffusion process is performed. Do the authors refer to this built-in VAE, or are they introducing an additional one? Furthermore, on which images was the VAE pretrained, and how many channels were used during pretraining?
> >
> > Conceptually, we refer to the VAE part and the diffusion module part of the latent diffusion pipeline separately. When we mentioned the diffusion model, we mean the module that only operates in the latent space of the VAE, and throughout the paper, we kept the VAE frozen.
> >
> > Therefore, the design components are: (i) how we utilize VAE to handle multi-channel and (ii) diffusion model design and training. Specifically, there is no additional VAE apart from a pretrained AutoencoderKL.
> >
> > We used "stabilityai/sd-vae-ft-mse" variant as the pretrained VAE, which is a finetuned version of the original “kl-f8 autoencoder” [8] on a 1:1 ratio of LAION-Aesthetics and LAION-Humans, an unreleased subset containing only SFW images of humans [9]. It is trained on natural images of 3 channels. As described in lines 146-161 we tackled the channel mismatch problem by stacking every individual channel 3 times, then processing through the frozen VAE, then concatenating the latents for every channel. Table 5 provides the reconstruction performance of the VAE on the RxRx1 dataset, which justifies our channel-stacking choice.
> >
> > > Previous methods [1,2] were already able to generate biologically meaningful images. What improvement does REPA provide in this case? Did the authors perform an ablation study to evaluate the importance of each component?
> >
> > [1] and [2] are image to image translation models. MorphGen, however, is purely a generative model that does not require an image conditioning to generate new samples. For the contribution of REPA, the reviewer can see the ablation on alignment in Table 6.
> >
> > > How is the conditioning performed? Which encoders are used to encode the different perturbations?
> >
> > For each perturbation id (also applies for cell type) we have a real-valued learnable embedding that represents the corresponding perturbation in a data driven way. The conditioning mechanism (described on lines 172-176) is achieved by a simple addition operation through the factors: timestep, perturbation and cell type embeddings.
> >
> > > FID and KID were originally proposed to evaluate RGB image generation. How do the authors apply these metrics to images with more than three channels?
> >
> > Following the literature in our comparisons [4, 5], we used the visualization tool (that performs a lossy conversion from a Cell Painting image to its RGB version) prior to FID/KID calculations. So the same post-processing is applied to both real and generated samples before taking the Inception features.
> >
> > > The U-Net used in modern diffusion models already includes self-attention blocks to capture spatial relationships. Does SiT provide any improvement over this? Did the authors compare the two backbones?
> >
> > We chose SiT based on prior work [6,7], which demonstrates that SiT/DiT variants achieve superior performance compared to UNet backbones by leveraging modern vision transformer architectures across multiple datasets and benchmarks.
> >
> > > I do not understand how Stable Diffusion was used as a baseline, since it does not include any encoder to handle perturbation conditioning. How is this achieved in the proposed setup? Furthermore, Stable Diffusion is trained on natural RGB images, so it seems unreasonable to apply it directly to biological images without retraining. Did the authors fine-tune or adapt the model in any way? What about the text encoder?
> >
> > We report the baseline introduced by [5] with the same name. What they did is using a similar conditioning mechanism (replacing the text encoder with learnable embeddings, but only for perturbation type, as they only use HUVEC cells) they finetuned the diffusion model (keeping VAE frozen) on the dataset.
> >
> > > In Table 1, was the number of channels the same for all methods? Although MorphenGen provides the best FID, the values remain very high. Furthermore, why did the authors not provide standard deviations for the other models?
> >
> > Yes the number of channels is the same, as we can only compute FID/KID using 3-channel images. MorphoDiff natively works in 3-channel as it applies the visualization tool as preprocessing, MorphGen works with original 6-channel images; therefore, upon generation, we used the same conversion to our samples. Since we matched the evaluation pipeline completely, we reported the numbers from [5] as is and we only provided CIs for our model.

---

> > > ### Author Response · Authors · 2025-11-19
> > >
> > > [1]  Bourou, A., Boyer, T., Gheisari, M., Daupin, K., Dubreuil, V., De Thonel, A., ... & Genovesio, A. (2024, January). PhenDiff: Revealing Subtle Phenotypes with Diffusion Models in Real Images. In MICCAI (3).
> > >
> > > [2] Lamiable, A., Champetier, T., Leonardi, F., Cohen, E., Sommer, P., Hardy, D., ... & Genovesio, A. (2023). Revealing invisible cell phenotypes with conditional generative modeling. Nature Communications, 14(1), 6386.
> > >
> > > [3] Palma, A., Theis, F. J., & Lotfollahi, M. (2025). Predicting cell morphological responses to perturbations using generative modeling. Nature Communications, 16(1), 505.
> > >
> > > [4] Zhang, Y., Su, Y., Wang, C., Li, T., Wefers, Z., Nirschl, J. J., ... & Yeung-Levy, S. CellFlux: Simulating Cellular Morphology Changes via Flow Matching. In Forty-second International Conference on Machine Learning.
> > >
> > > [5] Navidi, Z., Ma, J., Miglietta, E., Liu, L., Carpenter, A. E., Cimini, B. A., ... & WANG, B. MorphoDiff: Cellular Morphology Painting with Diffusion Models. In The Thirteenth International Conference on Learning Representations.
> > >
> > > [6] Peebles, W., & Xie, S. (2023). Scalable diffusion models with transformers. In Proceedings of the IEEE/CVF international conference on computer vision (pp. 4195-4205).
> > >
> > > [7] Ma, N., Goldstein, M., Albergo, M. S., Boffi, N. M., Vanden-Eijnden, E., & Xie, S. (2024, September). Sit: Exploring flow and diffusion-based generative models with scalable interpolant transformers. In European Conference on Computer Vision (pp. 23-40). Cham: Springer Nature Switzerland.
> > >
> > > [8] Rombach, R., Blattmann, A., Lorenz, D., Esser, P., & Ommer, B. (2021). High-Resolution Image Synthesis with Latent Diffusion Models. GitHub. https://github.com/CompVis/latent-diffusion
> > >
> > >
> > > [9] Stability AI. (2024, March 5). sd-vae-ft-mse. Hugging Face. Retrieved November 18, 2025, from https://huggingface.co/stabilityai/sd-vae-ft-mse

---

> ### Comment · Reviewer_g5zt · 2025-11-23
>
> We thank the authors for their responses and for providing additional results. While some of our initial concerns were addressed, several important issues remain, and new questions have emerged.
>
> * We respectfully disagree with the authors’ claim that REPA is novel in the way it is introduced. Applying REPA to biological imaging is not, in itself, a novelty. Moreover, previous work has already addressed **multi-channel bioimaging data [1]** as well as **conditioning generative models on different perturbations [2,3,4,5,6]**. Adding cell type as an additional conditional variable does not constitute methodological novelty; it simply increases the number of conditioning attributes. More generally, **we have no issue with adapting methods from other domains to biology if such adaptation yields new biological insights**. However, in the present case, the method does not introduce any significant technical contribution, and the results do not reveal new biological findings. As such, the overall **novelty remains very limited**. Achieving better FID or KID scores is not, by itself, a biological contribution.
>
>
> * We are also confused by the authors’ statement that their model is “purely generative and does not require a control sample to generate an image.” **All baseline methods discussed (MorphoDiff, CellFlux, IMPA) take a control image as input and generate a perturbed version**. If MorphoGen does not rely on control images, then the entire comparison to these baseline methods becomes inappropriate, as the task formulation is fundamentally different.
>
> * It is also essential to adhere to terminology established by the community. In particular, **latent diffusion models refer to architectures composed of a VAE coupled with a diffusion model**.
>
> * For Stable Diffusion, the authors did not train any model on their own dataset; **instead, they took results from [6] and inserted them into the paper without clearly disclosing this**. This practice is **highly unusual and may raise serious concerns**.
>
>
> We thank the authors again for their efforts in addressing the reviews, but in light of the remaining concerns outlined above, we will keep our original score unchanged.
>
>
>
> [1] Channel Vision Transformers: An Image Is Worth 1 x 16 x 16 Words, Bao et al.
>
> [2] Bourou, A., Boyer, T., Gheisari, M., Daupin, K., Dubreuil, V., De Thonel, A., ... & Genovesio, A. (2024, January). PhenDiff: Revealing Subtle Phenotypes with Diffusion Models in Real Images. In MICCAI (3).
>
> [3] Lamiable, A., Champetier, T., Leonardi, F., Cohen, E., Sommer, P., Hardy, D., ... & Genovesio, A. (2023). Revealing invisible cell phenotypes with conditional generative modeling. Nature Communications, 14(1), 6386.
>
> [4] Palma, A., Theis, F. J., & Lotfollahi, M. (2025). Predicting cell morphological responses to perturbations using generative modeling. Nature Communications, 16(1), 505.
>
> [5] Zhang, Y., Su, Y., Wang, C., Li, T., Wefers, Z., Nirschl, J. J., ... & Yeung-Levy, S. CellFlux: Simulating Cellular Morphology Changes via Flow Matching. In Forty-second International Conference on Machine Learning.
>
> [6] Navidi, Z., Ma, J., Miglietta, E., Liu, L., Carpenter, A. E., Cimini, B. A., ... & WANG, B. MorphoDiff: Cellular Morphology Painting with Diffusion Models. In The Thirteenth International Conference on Learning Representations.

---

> > ### Author Response · Authors · 2025-11-24
> >
> > > We respectfully disagree with the authors’ claim that REPA is novel in the way it is introduced. Applying REPA to biological imaging is not, in itself, a novelty. Moreover, previous work has already addressed multi-channel bioimaging data [1] as well as conditioning generative models on different perturbations [2,3,4,5,6]. Adding cell type as an additional conditional variable does not constitute methodological novelty; it simply increases the number of conditioning attributes. More generally, we have no issue with adapting methods from other domains to biology if such adaptation yields new biological insights. However, in the present case, the method does not introduce any significant technical contribution, and the results do not reveal new biological findings. As such, the overall novelty remains very limited. Achieving better FID or KID scores is not, by itself, a biological contribution.
> >
> > We have never claimed that applying REPA to biological imaging is novel. As stated ad verbatim in the reply above, our novelty concerns “incorporating **domain features into otherwise generic latent spaces** using a foundation model through representation alignment”. REPA had been shown to speed up training, **we found it also infuses biologically relevant signals** in the generated samples.
> >
> > Adding cell type as an additional conditional variable **not only** increases the number of conditioning attributes but **also allows compositional generalization (new perturbation/cell type combinations)**, as we discussed on Section 4.5.
> >
> > Moreover, we do achieve state-of-the-art FID/KID, but we do not stop there. We kindly refer the reviewer to
> >
> > - **Section 4.4** for **CellProfiler feature alignment** and **CATE consistency** between real and generated samples.
> > - **Section 4.5** for the **compositional generalization** results.
> > - **Appendix I and J** for our analysis on the **downstream classification performance** for the conditioning factors, spanning CellProfiler, OpenPhenom and ResNet as feature extractors.
> >
> >
> > Finally, just because other methods modeled conditioning generative models on different perturbations **does not imply** that the problem has been solved. Our paper establishes a new **state-of-the-art**, significantly outperforming the baselines in image quality ( ~35%  FID improvement) **in addition to the evaluation of biological features mentioned above**.
> >
> >
> > > We are also confused by the authors’ statement that their model is “purely generative and does not require a control sample to generate an image.” All baseline methods discussed (MorphoDiff, CellFlux, IMPA) take a control image as input and generate a perturbed version. If MorphoGen does not rely on control images, then the entire comparison to these baseline methods becomes inappropriate, as the task formulation is fundamentally different.
> >
> > The reviewer’s take is simply not true. **All the results in the main paper** compares **MorphGen** against **MorphoDiff** and **Stable Diffusion** baselines reported in [1] as **the task formulation is exactly the same**. CellFlux and IMPA are specifically mentioned in cross benchmark comparisons and are not included in the main paper. Specifically, we reported only Relative FIDs to give an insight on our performance across benchmarks.
> >
> >
> > > It is also essential to adhere to terminology established by the community. In particular, latent diffusion models refer to architectures composed of a VAE coupled with a diffusion model.
> >
> > Upon clarifications, if the reviewer still is unhappy with the terminology that is used, we can remove the word “latent”.
> >
> >
> > > For Stable Diffusion, the authors did not train any model on their own dataset; instead, they took results from [6] and inserted them into the paper without clearly disclosing this. This practice is highly unusual and may raise serious concerns.
> >
> > We follow **the same exact experimental protocol** as [1] did (given the released code by the authors). Given the fixed evaluation pipeline including the dataset, preprocessing, number of generated samples, real sample augmentations, the **target distribution between the models are the same** and therefore the **numbers are directly comparable**. We utilized the released code by [1] in our evaluation and we refer to the way we match the evaluations on **lines 248-258 and 701-709**.
> >
> > [1] Navidi, Z., Ma, J., Miglietta, E., Liu, L., Carpenter, A. E., Cimini, B. A., ... & WANG, B. MorphoDiff: Cellular Morphology Painting with Diffusion Models. In The Thirteenth International Conference on Learning Representations.

---

> > > ### Author Response · Authors · 2025-11-27
> > >
> > > We greatly appreciate your time and constructive comments. We trust that our responses have clarified your concerns and we remain available for any further clarification or discussion.

---

### Official Review · Reviewer_r9jV · 2025-10-31

**Soundness:** 3
**Presentation:** 2
**Contribution:** 2
**Rating:** 4
**Confidence:** 4

**Summary:**

This paper develops MorphGen, a diffusion model for generating cell painting images conditioned on a description of a perturbation and cell type. There are previous approaches for this task, but two key advantages of MorphGen are the ability to natively model all 6 channels of cell painting data and the ability to incorporate cell type conditions.

**Strengths:**

•	The work addresses an important problem—predicting perturbation response using cell painting data

•	Extending previous diffusion approaches to natively model the 6 cell painting channels is an important step toward more effective cell perturbation prediction models.

•	FID results are strong, and uncurated images look qualitatively realistic. The way this evaluation is conducted seems really solid.

•	Evaluations using CellProfiler features are important and show that the method captures biologically meaningful aspects of the images.

**Weaknesses:**

•	From an ML perspective, the work is more like an incremental step than a paradigm shift. This is a relatively standard diffusion model with a few tweaks to make it work on more channels than the natural images commonly used for training in computer vision applications.

•	From an applications perspective, it seems that out-of-sample prediction is not really evaluated (and maybe not possible with this approach, see next question). This is kind of the main goal of developing such a generative model in the first place.

•	Cell type and perturbation conditioning are not clearly described. How do you represent a chemical or genetic perturbation? Is it a one-hot encoding or the latent space of a chemical encoder? Using a chemical structure-based encoder of some kind seems like a better choice because it allows potential generalization to unseen perturbations.

•	Evaluations don’t really test whether the generated images respect the cell type or perturbation condition. Something like a conditional FID or classification accuracy on generated images would get at this more directly.

•	Important previous work not discussed: LUMIC, Hung et al. 2024. LUMIC uses a related latent diffusion approach, is designed to predict across cell types and can predict held-out perturbations and held-out cell types (though it does not predict all 6 channels like the current work).

**Questions:**

1. How do you represent a chemical or genetic perturbation? Is it a one-hot encoding or the latent space of a chemical/gene encoder?
2. How do you represent the cell type when conditioning the diffusion model?
3. Can the model in principle generalize to unseen perturbations or unseen cell types?

---

> ### Author Response · Authors · 2025-11-19
>
> We thank the reviewer for their careful and constructive evaluation. Below, we respond to each concern:
>
> >  From an ML perspective, the work is more like an incremental step than a paradigm shift.
>
> We respectfully disagree with the reviewer’s point in the context of our application of generative modeling to cellular microscopy, for the following reasons: .
>
> Compared to other methods in ML for CP image generation: previous methods only generate either low resolution  [1, 2] or lossy RGB images [3] (also ICLR2024 Oral) and study only a single cell type [1, 2, 3]. Instead, our model can address all these limitations while retaining a state-of-the-art performance. To do this, we proposed a simple yet effective technique, using the encoder to independently represent the different channels and concatenating them in the latent space.
>
> Compared to ML approaches in general: our choice to encode channels separately makes the latent dimension six times larger than the vanilla diffusion (used by MorphoDiff). This larger latent dimensionality clearly makes the diffusion process more difficult. Our surprising finding (see Table 6 for more details) is that representation alignment infuses morphological properties into the generated data, increasing biological fidelity. This is critical for downstream tasks like treatment effect estimation or simply property prediction, and was not demonstrated in the original paper.
>
>
> > From an applications perspective, it seems that out-of-sample prediction is not really evaluated. This is kind of the main goal of developing such a generative model in the first place.
>
> **Compositional generalization:** We provided a compositional generalization experiment on Section 4.5 (and for a detailed description see Appendix J), where we held out a pair of cell type and perturbation and trained the model on the remaining data. Therefore, the model has seen that perturbation with other cell types and that cell type with other perturbations. We believe compositional generalization is a natural next step that is underexplored, mainly because most models analyze only a single factor. We believe our paper opened the way to analyzing it by modeling multiple factors at the same time, which can be an important step towards using generative models for drug discovery.
>
> **Synthetic data generation for downstream tasks:** A generative model that represents in-distribution effectively can still be utilized in augmenting datasets (see Appendix H and I for a detailed analysis on downstream tasks), and the performance in the low-sample cases (data efficient) shows that it can potentially reduce the experiment costs (see Appendix G).
>
> > Cell type and perturbation conditioning are not clearly described. Using a chemical structure-based encoder of some kind seems like a better choice because it allows potential generalization to unseen perturbations.
>
> Lines 172-176 explain the conditioning mechanism. For each perturbation id (also applies for cell type), we have a real-valued learnable embedding that represents the corresponding perturbation in a data driven way. We agree that using a chemical structure-based encoder is a promising direction for modeling completely unseen scenarios. However, we deliberately evaluate the more realistic and testable scenario of compositional generalization: holding out a pairing of (cell type, perturbation) while each factor remains individually observed during training. In our view, compositional generalization is a natural next step before jumping into high-risk unseen cases.
>
> > Evaluations don’t really test whether the generated images respect the cell type or perturbation condition. Something like a conditional FID or classification accuracy on generated images would get at this more directly.
>
> The Tables 1 (conditioned on perturbation), 2 (conditioned on perturbation as well as organelle), 3 (conditioned on cell type) already report the conditional FID and KID. We explained the evaluation protocol in Section 4.1, however we will clarify the names and replace them with cFID and cKID respectively. Additionally, classification accuracies on generated images are already explained in detail in Appendix H and I. We utilized (i) OpenPhenom features, (ii) CellProfiler features and (iii) ResNet features from the generated images and reported their accuracies.
>
>
> > Important previous work not discussed: LUMIC, Hung et al. 2024.
>
> We will include a discussion about image to image translation models for drug discovery in our related work. However, we would like to note a key difference between LUMIC and MorphGen. LUMIC is a structure-aware style transfer model. It is designed to generate novel molecules **starting from a control image** with a structural embedding of the applied treatment. On the other hand, MorphGen is purely a generative model that offers rich phenotyping of known perturbations across many cell types and perturbations, as well as unseen combinations of the known factors.

---

> > ### Author Response · Authors · 2025-11-19
> >
> > [1] Palma, A., Theis, F. J., & Lotfollahi, M. (2025). Predicting cell morphological responses to perturbations using generative modeling. Nature Communications, 16(1), 505.
> >
> > [2] Zhang, Y., Su, Y., Wang, C., Li, T., Wefers, Z., Nirschl, J. J., ... & Yeung-Levy, S. CellFlux: Simulating Cellular Morphology Changes via Flow Matching. In Forty-second International Conference on Machine Learning.
> >
> > [3] Navidi, Z., Ma, J., Miglietta, E., Liu, L., Carpenter, A. E., Cimini, B. A., ... & WANG, B. MorphoDiff: Cellular Morphology Painting with Diffusion Models. In The Thirteenth International Conference on Learning Representations.

---

> > > ### Author Response · Authors · 2025-11-27
> > >
> > > We greatly appreciate your time and comments on our submission. We hope our responses addressed your questions and remain available for any further discussion.

---

### Official Review · Reviewer_Bc8w · 2025-10-31

**Soundness:** 3
**Presentation:** 3
**Contribution:** 3
**Rating:** 6
**Confidence:** 5

**Summary:**

This paper presents MorphGen, a diffusion-based generative model for Cell Painting microscopy images that achieves controllable generation across multiple cell types and perturbations. The key innovation is an alignment loss that guides MorphGen’s internal representations to match those of OpenPhenom (Kraus et al., 2024), a biological foundation model, encouraging the generative model to capture biologically meaningful features. Unlike prior works such as MorphoDiff (Navidi et al., 2025), which compressed six fluorescence channels into RGB and handled only one cell type, MorphGen generates all six channels jointly at higher resolution, thus preserving organelle-specific details essential for downstream morphological analysis. The model uses a latent diffusion architecture (leveraging a pretrained VAE) and incorporates conditioning on both cell type and perturbation. Experiments demonstrate that MorphGen produces morphologically plausible cell images that maintain known subcellular structures. Quantitatively, it significantly outperforms previous state-of-the-art: for example, on a multi-gene test set, its Fréchet Inception Distance (FID) is 35–60% lower than MorphoDiff. Qualitative results show that generated images closely mirror real cell images in texture and morphology. The paper also introduces evaluation metrics like Relative FID (normalised by dataset variability) and uses CellProfiler features to demonstrate that synthetic images capture phenotypic variation. Overall, the contributions of MorphGen are a substantial step toward “virtual cell” models for in silico biological experiments, enabling high-content image generation with controllable conditions and improved biological fidelity.

**Strengths:**

- Originality: The paper introduces a new combination of ideas focussed towards microscopy image analysis – diffusion models with a transformer backbone, multi-channel image generation, and alignment to a domain-specific foundation model. This is a creative extension of diffusion models into the biological imaging domain, addressing limitations of previous approaches. The representation alignment loss (adapted from REPA by Yu et al., 2025) is used in a novel way here (with OpenPhenom features) to inject biological priors into the generative process.

- Quality: The technical quality is high. The method is described in sufficient detail, and the experiments are decent; however could be better. The authors compare MorphGen against appropriate baselines (MorphoDiff and even Stable Diffusion repurposed) on multiple datasets. The quantitative gains are impressive. For instance, MorphGen achieves substantially lower FID/KID scores than MorphoDiff across datasets. The ablation studies (in the appendix) lend support that each component (alignment loss, full-channel generation, etc.) has a positive impact. The model outputs are of high resolution and fidelity; Figure 2 and others show that synthetic images reproduce fine subcellular details, which is non-trivial. Additionally, the paper reports not only generative quality metrics but also uses CellProfiler features and a CATE (conditional treatment effect) analysis to ensure that known phenotypic differences under perturbations are being captured – this indicates a quality focus on biological accuracy, not just visual fidelity.

- Clarity: Aside from minor issues noted, the paper is clearly written.

- Significance: This work has practical significance for biomedical imaging communities. By enabling controllable simulation of cell images, MorphGen can be used to generate in silico experiments – for example, creating hypothetical outcomes for perturbations or augmenting datasets for training. The ability to model multiple cell types and stains is particularly significant, as it broadens the applicability (previous models were limited in scope).

**Weaknesses:**

While the paper is strong, there are some weaknesses and areas for improvement:
- Evaluation could be more biologically insightful: The current evaluation leans on aggregate metrics (FID, KID) and visual inspection, with some PCA and correlation analyses in the appendix. However, these don’t fully demonstrate that the generated images recapitulate known biological relationships. For instance, a more direct test would be to see if specific CellProfiler features correlate between real and generated cells from the same perturbation. The paper shows side-by-side PCA of real vs fake and a global correlation matrix, but this is only a coarse validation. It would strengthen the work to quantify, for example, that for each known perturbation, the change in particular CellProfiler features (nuclear size, cell count, etc.) in generated images correlates with that in real images. Moreover, the authors could calculate the recall of known biological relationships between genes based on databases like StringDB, and compare this score between real and generated images. See Celik et al. 2024 (https://doi.org/10.1371/journal.pcbi.1012463).  In short, demonstrating downstream task fidelity (such as predicting drug mechanism or gene function from synthetic images and comparing to real) would make the biological validity more convincing.


- Limited discussion of foundation model choice: The authors use OpenPhenom embeddings to guide the generator. OpenPhenom is a reasonable choice (a well-known cell image foundation model), but the paper doesn’t explore this decision deeply. One concern is that recent analyses suggest such foundation models may be dominated by easy-to-learn signals like cell count (how many cells in the image) rather than subtler phenotypes. If OpenPhenom’s embedding primarily captures cell count or other simple variations, aligning to it might inadvertently make MorphGen focus on those and neglect finer morphological details. There are other biological feature embedding models they could consider – for example, CellCLIP (Lu et al., 2025) aligns Cell Painting images with text descriptions of perturbations via contrastive learning, MolPhenix (Fradkin et al., 2024) aligns images with molecular structures, CLOOME (Sanchez-Fernandez et al., 2023) is a confounder-aware multimodal model linking cell images and chemicals, CWA-MSN learns representations via siamese networks. All of the above provide pre-trained image embedders for cell painting images that have outperformed OpenPhenom in recalling known biological relationships from images. An ablation or comparison using some of these different embeddings (or simply turning off the alignment loss) would reveal how crucial the choice is. It’s possible that OpenPhenom is not uniquely optimal and that other representations might improve or alter the results. Currently, the paper assumes OpenPhenom as a given; examining this would improve the work’s robustness and novelty.



- Clarity and definition issues: There are a few spots where the paper could be clearer. Terms like IMPA should be defined when first used. Not defining it might confuse readers unfamiliar with that prior work. Similarly, “clean images” is used in line 200. Does this mean images without noise? The authors should specify this to avoid ambiguity. The notation $z_0$ appears without definition (likely the initial noise latent for diffusion sampling), as does F(x) (I assume OpenPhenom). Explicitly stating this would help readers follow the generation process description. Furthermore, Scalable Interpolant Transformer (SiT) is defines twice in lines 163 and 185. These are relatively small weaknesses, but improving them would polish the paper.


- Use of a pretrained VAE not specific to microscopy: The model relies on a pretrained VAE to encode and decode images. This VAE was originally trained on RGB natural images. The authors adapt it for 6-channel input by stacking channels into pseudo-RGB triplets, which is clever. However, the paper does not mention any fine-tuning of this VAE on cell images. Using a VAE not trained on fluorescent microscopy data could introduce a domain gap – e.g., color/intensity distributions in natural images differ from microscopy, and the VAE might not optimally compress/reconstruct cell structures (especially if cell images violate assumptions it learned). It’s a testament to the method that it still works well, but this choice could be a limitation. Perhaps training a custom VAE on Cell Painting (even a smaller one) might further improve quality. At minimum, the authors should clarify what data the VAE was pretrained on and discuss any limitations or justify why this doesn’t harm results. Right now, it’s a bit implicit.


- Miscellaneous: I have a few other minor critiques. (1) The paper uses “interpretability” in describing the benefits of full-channel generation. While preserving organelle channels does aid human interpretability of results, the model itself isn’t inherently interpretable in a model-explainability sense. It’s more about facilitating post-hoc analysis. The wording could be tempered to avoid overstating interpretability. (2) The comparison to CellFlux (another recent generative model, possibly via flow matching) is only mentioned briefly in the appendix. If CellFlux is contemporary work, a clearer comparison in the main text would be helpful for completeness. These issues do not fundamentally weaken the work but addressing them would improve the overall presentation and rigour.

**Questions:**

- Foundation model alignment: Can the authors provide more insight into the decision to use OpenPhenom embeddings and how sensitive the results are to this choice? An ablation on at least one another cell painting image embedder would be appreciated. For example, if one trains MorphGen without the alignment loss (or with a different embedding space, like CellCLIP), how does the image quality or biological fidelity change?
- Scope of “biologically meaningful features”: The paper claims that the alignment enforces capturing meaningful patterns. Could the authors elaborate on which phenotypic patterns MorphGen is actually learning? In short, how do we know the model isn’t just learning to generate generic-looking cells plus the correct number of cells, rather than truly phenotype-specific morphologies? Any additional evidence here would strengthen confidence in biological relevance. A comparison between using real and generated images to recall known biological relationships from rxrx3-core would make this paper much stronger.

---

> ### Author Response · Authors · 2025-11-19
>
> We thank the reviewer for their thoughtful and constructive feedback. Below, we address the main points raised and summarize the clarifications and improvements made in the revised manuscript.
>
> >  Evaluation could be more biologically insightful.
>
> As requested, we evaluated biological fidelity by comparing the change in CellProfiler features between real and generated cells across three perturbations (1138 control, 1108, 1124, 1137). For each feature, we computed the mean difference between each treatment and the control (delta-feature) separately for real and generated images, and then we computed the correlation between these delta-values. This captures whether the generator preserves both the direction and magnitude of perturbation-induced morphological changes. The resulting correlations are high for biologically meaningful texture and shape features (not only global properties such as cell count), while noisy or non-biological features show lower agreement, indicating that the model reproduces true phenotypic effects rather than artifacts.
>
> Top 10 Features by Correlation
> | feature                                               | correlation | p_value  | mean_abs_delta_real | mean_abs_delta_gen |
> |-------------------------------------------------------|-------------|----------|----------------------|---------------------|
> | StDev_Cytoplasm_Texture_Correlation_OrigGray_5_03_256 | 0.999914    | 0.008358 | 0.615383             | 0.385262            |
> | StDev_Cells_AreaShape_FormFactor                      | 0.999843    | 0.011268 | 0.550508             | 0.436651            |
> | Mean_Cells_AreaShape_Area                             | 0.999537    | 0.019373 | 0.603212             | 0.672867            |
> | StDev_Nuclei_Texture_Correlation_OrigGray_10_00_256   | 0.999419    | 0.021701 | 0.730174             | 0.649439            |
> | AreaOccupied_AreaOccupied_Cells                       | 0.999331    | 0.023281 | 1.000485             | 0.959291            |
> | StDev_Nuclei_Intensity_MassDisplacement_OrigGray      | 0.999148    | 0.026276 | 0.468416             | 0.386986            |
> | Granularity_8_OrigGray                                | 0.999077    | 0.027356 | 0.271900             | 0.391171            |
> | StDev_Cells_Granularity_8_OrigGray                    | 0.999036    | 0.027954 | 0.438343             | 0.456063            |
> | ImageQuality_StdIntensity_OrigGray                    | 0.998786    | 0.031377 | 0.687281             | 0.684044            |
> | StDev_Cells_AreaShape_BoundingBoxMaximum_Y            | 0.998211    | 0.038081 | 0.341598             | 0.258343            |
>
>
>
> Bottom 10 Features by Correlation
> | feature                                            | correlation | p_value  | mean_abs_delta_real | mean_abs_delta_gen |
> |----------------------------------------------------|-------------|----------|----------------------|---------------------|
> | Mean_Nuclei_Texture_Correlation_OrigGray_10_02_256 | 0.575836    | 0.609354 | 0.688423             | 0.643494            |
> | Mean_Nuclei_AreaShape_BoundingBoxArea              | 0.547925    | 0.630836 | 0.640173             | 0.306950            |
> | StDev_Cells_AreaShape_Zernike_2_0                  | 0.530413    | 0.644074 | 0.829925             | 0.540368            |
> | Mean_Nuclei_AreaShape_Zernike_2_2                  | 0.512267    | 0.657611 | 0.685602             | 0.196343            |
> | Mean_Cells_AreaShape_Eccentricity                  | 0.366334    | 0.761226 | 1.253528             | 1.004303            |
> | Granularity_13_OrigGray                            | 0.235079    | 0.848931 | 0.682751             | 0.435022            |
> | Median_Nuclei_Granularity_15_OrigGray              | -0.395548   | 0.741110 | 0.135087             | 0.164899            |
> | ExecutionTime_24OverlayOutlines                    | -0.426113   | 0.719765 | 0.391609             | 0.364377            |
> | ExecutionTime_23MeasureImageIntensity              | -0.626717   | 0.568796 | 0.345231             | 0.294323            |
> | ExecutionTime_06MeasureImageQuality                | -0.989645   | 0.091696 | 0.355497             | 0.256092            |
>
>
> - **Mean Correlation:** 0.7942
> - **Median Correlation:** 0.9531
> - **Features with correlation > 0.8:** 51
> - **Features with correlation > 0.5:** 64
> - **Features with p-value < 0.05:** 12

---

> > ### Author Response · Authors · 2025-11-19
> >
> > We also quantified biological faithfulness at the perturbation level. For each perturbation, we computed delta-feature vectors across 70 CellProfiler features for both real and generated images, and measured the Pearson correlation between these two vectors. This evaluates whether the generator reproduces the overall phenotypic signature of each perturbation, rather than individual features. Similarly, all three perturbations showed strong agreement.
> >
> > | perturbation_id | correlation |       p_value       | n_features | n_samples_real | n_samples_gen | mean_abs_delta_real | mean_abs_delta_gen |
> > |-----------------|-------------|----------------------|------------|----------------|----------------|----------------------|---------------------|
> > | 1137            | 0.922121    | 9.441077e-30         | 70         | 126            | 233            | 0.677079             | 0.570834            |
> > | 1108            | 0.895998    | 1.137949e-25         | 70         | 130            | 250            | 0.349373             | 0.288803            |
> > | 1124            | 0.834478    | 2.887708e-19         | 70         | 127            | 243            | 0.459971             | 0.459371            |
> >
> >
> >
> > Additionally, we now also report CATE (conditional average treatment effect) using CellProfiler features.
> >
> > | comparison        | ATE_real  | ATE_generated | ATE_difference | ATE_ratio |
> > |-------------------|-----------|---------------|----------------|-----------|
> > | p1138_vs_p1108    | 13.340911 | 10.658878     | 2.682033       | 0.798962  |
> > | p1138_vs_p1124    | 22.383785 | 21.644595     | 0.739190       | 0.966977  |
> > | p1138_vs_p1137    | 46.801704 | 35.193803     | 11.607901      | 0.751977  |
> >
> > Finally, we would like to note that, to validate the biological faithfulness, in addition to CATE, we evaluated our model on a downstream classification task on Appendix H to predict the conditioning factors (perturbation type, and cell type) using OpenPhenom, CellProfiler and ResNet features to demonstrate the agreements and the quality in an unbiased way. We hope that these new experiments and clarifications address the reviewer’s concerns.
> >
> >
> > > Limited discussion of foundation model choice.
> >
> > We thank the reviewer for their comments. We agree that, in addition to aligned vs. not aligned models, exploring other foundation models to align and demonstrate consistent gains would make the paper stronger. Training a diffusion model from scratch is computationally expensive, so we selected CellCLIP as a representative alternative and present our results below:
> >
> > | Method | FID ↓ | KID ↓ |
> > |---------------------------|-----------------|--------------------|
> > | MorphGen w/o align. | 56.87 ± 3.35 | 0.023 ± 0.001 |
> > | MorphGen w/ CellCLIP | 53.70 ± 2.11 | 0.022 ± 0.002 |
> > | MorphGen w/ OpenPhenom | 50.20 ± 2.45 | 0.018 ± 0.000 |
> >
> > The table shows that aligning to CellCLIP also improves over the non-aligned baselines, indicating the benefit of representation alignment is not specific to OpenPhenom.
> >
> >
> > > Use of a pretrained VAE not specific to microscopy
> >
> > Yes, the pretrained VAE is trained on natural images, however Table 5 presents its performance on the RxRx1 images. With the channel stacking, the pretrained VAE achieves an MSE on the scale of 4e-5, which is enough to justify our choice. While low reconstruction error is not sufficient on its own, it is a necessary condition for latent diffusion to be meaningful; with our channel stacking, the chosen VAE satisfies this requirement. A more powerful model trained with the samples from the biology domain would strengthen the results but considering that the current version already has a very small reconstruction error, the costs would clearly outweigh the benefits.
> >
> >
> > > Clarity and definition issues.
> >
> > We completely agree with all the points the reviewer has raised and appreciate their attention. We defined IMPA on its first appearance, avoided multiple definitions of SiT. We also fixed unnecessary stress on “clean image”. Finally we added a brief explanation to clarify F is OpenPhenom. You can see the updated manuscript (changes are highlighted with blue).
> >
> >
> > > Miscellaneous: Interpretability and CellFlux.
> >
> > We tempered the interpretability claims by replacing them with post-generation interpretability. CellFlux is briefly discussed in the related work, and detailed comparison is left to the Appendix F. The reason why we chose that is simply because CellFlux follows a fundamentally different evaluation pipeline from ours (native full channel 512x512 on all four cell types vs 96x96 U2OS crops), and explaining the way we matched evaluations would be very hard to fit within the main-paper space.

---

> > > ### Author Response · Authors · 2025-11-27
> > >
> > > We greatly appreciate your time and constructive comments. We kindly ask the reviewer if their questions were answered to their satisfaction and remain available for any further discussion.

---

### Author Response · Authors · 2025-12-02

We sincerely appreciate the time and effort invested by the AC and the reviewers in assessing our submission. In light of the current review process, we would like to briefly summarize where the paper stands after the discussion and revisions, focusing on the main substantive issues raised by the reviewers and our additions to strengthen the paper.

---

- **Biological faithfulness and aligning other models (Reviewer Bc8w).**

In response to Bc8w, we substantially expanded our biological evaluation. We now provide additional evidence of biological faithfulness by:

&nbsp;&nbsp;&nbsp;&nbsp;- Quantifying correlations between real vs. generated perturbation-induced changes in CellProfiler features, both at the feature and perturbation level

&nbsp;&nbsp;&nbsp;&nbsp;- Reporting CATE (conditional average treatment effect) consistency between real and generated images using CellProfiler features


&nbsp;&nbsp;&nbsp;&nbsp;- Exploring CellCLIP to align in addition to OpenPhenom alignment to demonstrate the consistent gains

---

- **Purely generative setting vs. image-to-image translation (Reviewers r9jV, g5zt).**

By adding the schematic (Fig. 6), we clarified that MorphGen, like MorphoDiff, is a *generative* model and does **not** require a control image as input. Image2image translation methods (e.g., LUMIC, PhenDiff, IMPA) instead map a control image to a perturbed one and therefore address a different task. All our main results compared MorphGen against MorphoDiff and Stable Diffusion baselines while matching the exact same evaluation and keeping the target distribution the same.

We stress that our evaluations: CellProfiler analyses, CATE, compositional generalization and downstream classification go beyond just perturbation and cell-type conditioned FID/KID.

---

- **Compositional generalization rather than perturbation encoders (Reviewers r9jV, g5zt).**

Unlike approaches that rely on pretrained perturbation encoders to extrapolate to new compounds, our generalization is achieved via compositionality over multiple conditioning factors. MorphGen is jointly conditioned on cell type and perturbation with learnable embeddings, enabling evaluation on **unseen (cell type, perturbation) pairs** while each factor remains observed individually during training. This multi-factor setup is, to our knowledge, new in this domain and is explicitly tested in our compositional generalization experiment.

---

- **Lack of novelty / incremental comments (Reviewers r9jV, g5zt).**

Our contribution is in *incorporating domain-specific foundation model features into otherwise generic latent spaces via representation alignment* to improve both biological faithfulness and generation quality. We use the REPA loss as a mechanism to perform this alignment and in our setting we find that it injects biologically relevant signals into the generated samples, as supported by the ablation with CellCLIP alignment, as well as our CellProfiler, CATE, and downstream analyses.

---

We hope this concise summary helps in forming an independent assessment of the submission. Thank you again for your time and effort in overseeing this process.

---

### Meta-Review · Area_Chair_NGHK · 2026-01-07

**Summary:**

The paper propose a generative model for cell imaging. The novelty of the approach as stated by the reviewers is limited as well as the comparison to other works. The practice of comparing done in the paper is also questionable as mentioned by reviewer g5zt. I think the comparison in the paper should be reconsidered. The paper claims that their measures are better for the biological problem at hand. Given that, a journal in that field that appreciates such measures might be a better fit. It might be that for biologists there is a great impact here despite the limited algorithmic novelty, but ICLR might not be the right venue for the paper. For an ML venue, the algorithm part and evaluation should be improved, or alternatively target a biological audience which might be a better fit.

**Reviewer Concerns:**

I don't think that the concerns of the reviewers have been met. It feels like the authors mainly tried to justify their claims without meeting the critic raised which I also share. Therefore, the paper cannot be accepted.

**Reviewer Scores:**

I think the reviewers' concerns were definitely not met. They even became worse. I would even suspect that g5zt would reduce further the score as the concerns raised were unmet.

---

### Decision · Program_Chairs · 2026-01-26

Reject